October 7, 2022

# Retraction Notice

Retraction: Following an investigation by PeerJ Computer Science the authors are in agreement that content in this publication (namely figures and concepts) was inappropriately reused from other published work (Wang, 2020). In consultation with PeerJ Computer Science, the authors request that the article be retracted.

PeerJ Editorial Office. 2022. Retraction: A comprehensive review of deep learning-based single image super-resolution. PeerJ Computer Science 8:e621 https://doi.org/10.7717/peerj-cs.621/retraction



# A comprehensive review of deep learning-based single image super-resolution

Syed Muhammad Arsalan Bashir[1,2,*], Yi Wang[1,*], Mahrukh Khan[3] and Yilong Niu[4]

[1] School of Electronics and Information, Northwestern Polytechnical University, Xi'an, Shaanxi, China
[2] Quality Assurance, Pakistan Space and Upper Atmosphere Research Commission, Karachi, Sindh, Pakistan
[3] Department of Computer Science, National University of Computer and Emerging Sciences, Karachi, Sindh, Pakistan
[4] School of Marine Science and Technology, Northwestern Polytechnical University, Xi'an, Shaanxi, China
* These authors contributed equally to this work.

Corresponding authors
Syed Muhammad Arsalan Bashir,
smarsalan@mail.nwpu.edu.cn
Yilong Niu,
yilong_niu@nwpu.edu.cn

## ABSTRACT

Image super-resolution (SR) is one of the vital image processing methods that improve the resolution of an image in the field of computer vision. In the last two decades, significant progress has been made in the field of super-resolution, especially by utilizing deep learning methods. This survey is an effort to provide a detailed survey of recent progress in single-image super-resolution in the perspective of deep learning while also informing about the initial classical methods used for image super-resolution. The survey classifies the image SR methods into four categories, i.e., classical methods, supervised learning-based methods, unsupervised learning-based methods, and domain-specific SR methods. We also introduce the problem of SR to provide intuition about image quality metrics, available reference datasets, and SR challenges. Deep learning-based approaches of SR are evaluated using a reference dataset. Some of the reviewed state-of-the-art image SR methods include the enhanced deep SR network (EDSR), cycle-in-cycle GAN (CinCGAN), multiscale residual network (MSRN), meta residual dense network (Meta-RDN), recurrent back-projection network (RBPN), second-order attention network (SAN), SR feedback network (SRFBN) and the wavelet-based residual attention network (WRAN). Finally, this survey is concluded with future directions and trends in SR and open problems in SR to be addressed by the researchers.

## INTRODUCTION

The image-based computer graphics models lack resolution independence (*Freeman, Jones & Pasztor, 2002*) as the images cannot be zoomed beyond the image sample resolution without compromising the quality of images. This is the case, especially in realistic images, for instance, natural photographs. Thus, simple image interpolation will lead to the blurring of features and edges within a sample image.

The concept of super-resolution was first used by *Gerchberg (1974)* to improve the resolution of an optical system beyond the diffraction limit. In the past two decades, the concept of super-resolution (SR) is defined as the method of producing high-resolution (HR) images from a corresponding low-resolution (LR) image. Initially, this technique was classified as spatial resolution enhancement (*Tsai & Huang, 1984*). The applications of super-resolution include computer graphics (*Kim, Lee & Lee, 2016a,b*; *Tao et al., 2017*), medical imaging (*Bates et al., 2007*; *Fernández-Suárez & Ting, 2008*; *Huang et al., 2008*; *Hamaide et al., 2017*; *Jurek et al., 2020*; *Teh et al., 2020*; *Bashir & Wang, 2021a*), security, and surveillance (*Zhang et al., 2010*; *Shamsolmoali et al., 2018*; *Lee, Kim & Heo, 2020*), which shows the importance of this topic in recent years.

Although being explored for decades, image super-resolution remains a challenging task in computer vision. This problem is fundamentally ill-posed because there can be several HR images with slight variations in camera angle, color, brightness, and other variables for any given LR image. Furthermore, there are fundamental uncertainties among the LR and HR data since the downsampling of different HR images may lead to a similar LR image, making this conversion a many-to-one process (*Yang & Yang, 2013*).

The existing methods of image super-resolution can be categorized into single-image super-resolution (SISR) and multiple-image approaches. In single image SR, the learning is performed for single LR-HR pair for a single image, while in multiple-image SR, the learning is performed for a large number of LR-HR pairs for a particular scene, thereby enabling the generation of an HR image from a scene (multiple images) (*Kawulok et al., 2020*). Video super-resolution deals with multiple successive images (frames) and utilizes the relationship within the frames to super-resolve a target frame; it is a special type of multiple image SR where the images are part of a scene containing different frames (*Liu et al., 2020b*).

In the past, classical SR methods such as statistical methods, prediction-based methods, patch-based methods, edge-based, and sparse representation methods were used to achieve super-resolution. However, recently the advances in computational power and big data have made researchers use deep learning (DL) to address the problem of SR. In the past decade, deep learning-based SR studies have reported superior performance than the classical methods, and DL methods have been used frequently to achieve SR. Researchers have used a range of methods to explore SR, ranging from the first method of Convolutional Neural Network (CNN) (*Dong et al., 2014*) to the recently used Generative Adversarial Nets (GAN) (*Ledig et al., 2017*). In principle, the methods used in deep learning-based SR methods vary in hyper-parameters such as network architecture, learning strategies, activation functions, and loss functions.

In this study, a brief overview of the classical methods of SR is outlined initially, whereas the main focus is given to give an overview of the most recent research in SR using deep learning. Previous studies have explored the literature on SR, but most of these studies emphasize the classical methods (*Borman & Stevenson, 1998*; *Park, Park & Kang, 2003*; *Van Ouwerkerk, 2006*; *Yang, Ma & Yang, 2014*; *Thapa et al., 2016*), additionally (*Yang, Ma & Yang, 2014*; *Thapa et al., 2016*) used human visual perception to gauge the performance of SR methods.

In recent years, there have been some reviews (*Ha et al., 2019*; *Yang et al., 2019*; *Zhang et al., 2019c*; *Zhou & Feng, 2019*; *Li et al., 2020*) focused on deep learning-based image super-resolution. The study by *Yang et al. (2019)* was focused on the deep learning methods for single image super-resolution. *Zhang et al. (2019c)* limited the scope of image SR to CNN-based methods for space applications, thereby only reviewing four methods namely, SRCNN, FSRCNN, VDSR and DRCN. *Ha et al. (2019)* reviewed the state-of-the-art SISR methods and classified them based on the type of framework, i.e., CNN, RNN-CNN-based methods and GAN-based methods. *Zhou & Feng (2019)* briefly reviewed some of the state-of-the-art SISR methods and provided an introduction of some of the methods without any evaluation of comparison of methods, while *Li et al. (2020)* reviewed the state-of-the-art methods in image SR while emphasizing on the methods based on CNNs and GANs for real-time applications. These review papers did not encompass the domain of super-resolution as a whole, and this paper fills that research gap by providing an overview of both classical and deep learning-based methods. At the same time, we have reviewed the deep learning-based methods into subdomain based on the functional blocks, i.e., upsampling methods, SR networks, learning strategies, SR framework and other improvements. This review paper fills the gap of a comprehensive review where a reader could access the overall progress of image super-resolution with appropriate section for the overall image quality metrics, SR methods, datasets, applications, and challenges in the field of image SR.

This survey is a comprehensive overview of the recent advances in SR, emphasizing deep learning-based approaches and their achievements in systematically achieving SR. Tables S1 and S2 respectively show the complete list of symbols and acronyms used in this study.

The key features of this study are:

1. We highlight the brief overview of the classical methods in SR and their contributions in light of past studies to give perspective.

2. We provide a detailed survey of deep learning-based SR, including the definition of the problem, dataset details, performance evaluation, deep learning methods used for SR, and specific applications where these SR methods were used and their performance.

3. We compare and contrast the recent advances in deep learning-based SR methods by summarizing the bounds of the methods by providing details of components of the SR methods used structurally.

4. Finally, the open problems in SR and critical challenges that require further probing are highlighted in this survey to provide future directions in SR.

This study is organized as follows:

In "Introduction", we have introduced the concept of SR and the overall overview of this study. In Fig. 1, we have summarized the hierarchical structure of this review. There are four main sections: classical methods, deep learning-based methods, applications of SR, Discussion, and future directions. In "Super-Resolution: Definitions and Terminologies", we put forward the problem definition and details of the evaluation dataset. "Survey

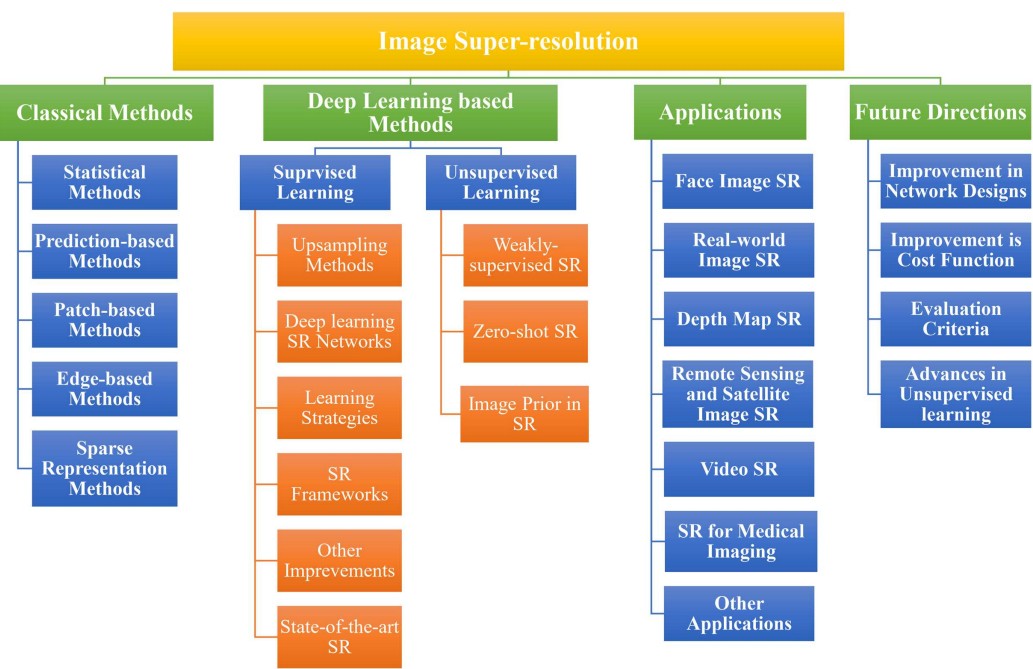

**Figure 1 Hierarchical classification of this survey.** Four main categories are (a) classical methods of image super-resolution, (b) deep learning-based methods for SR, (c) applications of super-resolution, (d) future research and directions in SR. *Green* color represent first-level sections, the blue color is for second-level subsections, and orange color represent third level subsections.

Methodology" discusses the methodology for the selection of studies included within this review. In "Conventional Methods of Super-Resolution", we compare and contrast the classical methods of SR, whereas, in "Supervised Super-Resolution", the SR methods based on supervised deep learning are explored. "Unsupervised Super-Resolution" covers the studies that used unsupervised deep learning-based methods for SR, and in "Domain-Specific Applications of Super-Resolution", various field-specific applications of SR in recent years are discussed. "Discussion and Future Directions" summarizes open challenges and limitations in current SR methods and puts forward future research directions, while "Conclusion" highlights the conclusions.

## SUPER-RESOLUTION: DEFINITIONS AND TERMINOLOGIES

In this section, the problem definition and the associated concepts of image super-resolution are discussed in light of the literature review.

### Single image super-resolution—problem definition

The image SR focuses on the recovery of an HR image from LR image input as and in principle, the LR image $I_{xLR}$ can be represented as the output of the degradation function, as shown in (1).

$$I_{xLR} = d(I_{yHR}, \partial) \tag{1}$$

Where $d$ is the SR degradation function that is responsible for the conversion of HR image to LR image, $I_{yHR}$ is the input HR image (reference image), whereas $\partial$ depicts the input parameters of the image degradation function. Degradation parameters are usually scaling factor, blur type, and noise. In practice, the degradation process and dependent parameters are unknown, and only LR images are used to get HR images by the SR method. The SR process is responsible for predicting the inverse of the degradation function d, such that $g = d^{-1}$

$$g(I_{xLR}, \delta) = d^{-1}(I_{xHR}) = I_{yE} \approx I_{yHR} \tag{2}$$

Where $g$ is the SR function, $\delta$ depicts the input parameters to the function $g$, and $I_{yE}$ is the estimated HR corresponding to the input $I_{xLR}$ image. It is also worth noticing that the super-resolution function, as in (2), is ill-posed, as the function $g$ is a non-injective function; thus, there are infinite possibilities of $I_{yE}$ for which the condition $d(I_{yE}, \partial) = I_{xLR}$ will hold.

The degradation process for the input LR images is unknown, and this process is affected by numerous factors such as sensor-induced noise, artifacts created because of lossy compression, speckle noise, motion blur, and misfocused images. In the literature, most of the studies have used a single downsampling function as the image degradation function:

$$d(I_{yHR}, \partial) = (I_{yHR})\downarrow_{s_f}, \{s\} \subseteq \partial \tag{3}$$

Where $\downarrow_{s_f}$ is the downsampling operator with $s_f$ being the scaling factor. One of the frequently used downsampling functions in SR is the bicubic interpolation (*Shi et al., 2016*; *Zhang & An, 2017*; *Shocher, Cohen & Irani, 2018*) with antialiasing. In some studies, like (*Zhang, Zuo & Zhang, 2018*), researchers have used more operations in the downsampling function, and the overall downsampling operation is:

$$d(I_{yHR}, \partial) = (I_{yHR} \otimes \kappa)\downarrow_{s_f} + n_\sigma, \{\kappa, s, \sigma\} \subseteq \partial \tag{4}$$

Where $I_{yHR} \otimes \kappa$ depicts the convolution of the HR image $I_{yHR}$ with the blurring kernel $\kappa$, $n_\sigma$ represents the additive white Gaussian noise with a standard deviation of $\sigma$. The degradation function defined in (4) and Fig. 2 is closer to the actual function as it considers more parameters than the simple downsampling degradation function (*Zhang, Zuo & Zhang, 2018*).

Finally, the purpose of SR is to minimize the loss function as follows:

$$\hat{\phi} = \left[\min \mathcal{L}(I_{yE}, I_{yHR})\right]_\phi + h\Psi(\phi) \tag{5}$$

Where $\mathcal{L}(I_{yE}, I_{yHR})$ is the loss function between the output HR image of SR and the actual HR image, $h$ is the tradeoff parameter, whereas $\Psi(\phi)$ is the regularization term. The most common loss function used in SR is the pixel-based mean square error (MSE), which can also be referred to as pixel loss. In recent years, researchers have used a combination of various loss functions, and these combinations are further explored in later

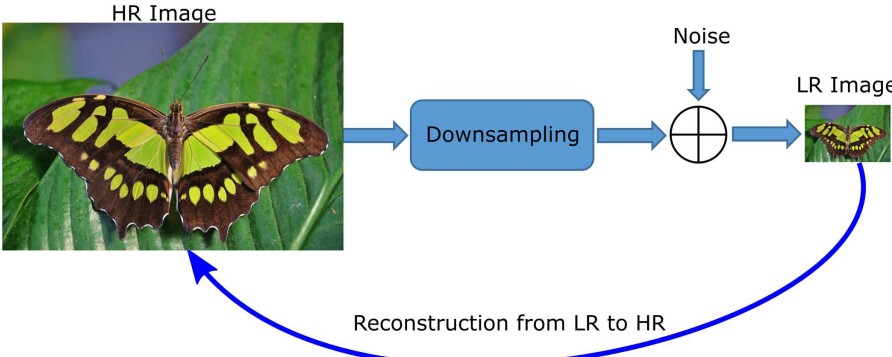

**Figure 2 Downsampling and upsampling in super-resolution.** Noise is added to simulate realistic degradation within an image.

sections. Further mathematical modeling of the SR problem is discussed in *Candès & Fernandez-Granda (2014)*.

## Methods for quality of SR images

Image quality can have several definitions as per the measurement methods, and it is generally a measure of the quality of visual attributes and perception of the viewers. The image quality assessment (IQA) methods are characterized into subjective methods (human perception of an image is natural and of good quality) and objective methods (quantitative methods by which image quality can be numerically computed) (*Thung & Raveendran, 2009*).

Quality-related visual aspects of an image are mostly a good measure, but this method requires more resources, especially if the dataset is large (*Wei, Yuan & Cai, 1999*); thus, in SR and computer vision tasks, the more suitable methods are objective. As per (*Saad, Bovik & Charrier, 2012*), the IQA methods are primarily categorized into three categories, i.e., reference image-based features from the actual image and blind IQA with no information about the ground truth. In this section, IQA methods primarily used in the domain of SR are further explored.

### Peak signal-to-noise ratio

In information systems, the peak signal-to-noise ratio (PSNR) is a measurement technique for analyzing the signal power compared to the noise power, especially in images; the PSNR is used as a quantitative measure of the compression quality of an image. In super-resolution, the PSNR of an image is defined by the maximum pixel value and the mean square error between the reference image and the SR image, also known as the power of image distortion noise. For a given maximum pixel value ($M$) and the reference image ($I_r$) having t pixels and the SR image ($I_y$), the peak signal-to-noise ratio is defined as:

$$PSNR = 10\log_{10}\left(\frac{M^2}{MSE}\right) \tag{6}$$

Where M is mostly for 8-bit color space depth, i.e., the max value of 255 and *MSE* is given by:

$$MSE = \frac{1}{t} \sum_{i=1}^{t} \left( I_r(i) - I_y(i) \right)^2 \tag{7}$$

As seen from (6), the PSNR is related to the individual pixel intensity values of the SR image and reference image and is a pixel-based metric of image quality. In some cases (*Almohammad & Ghinea, 2010*; *Horé & Ziou, 2010*; *Goyal, Lather & Lather, 2015*), this quality metric can be misleading as the overall image might not be visually similar to that of the reference image. This metric is still used for image comparisons, especially comparing the results of SR algorithms with previously published results to compare the working of any new method in the field of SR.

MSE for color images averaged for color channels, and an alternate approach is to measure PSNR for luminance and or greyscale channels separately as the human eye is more sensitive to changes in luminance in contrast to changes in chrominance (*Dabov et al., 2006*).

### Structural similarity index

The visual perception of humans is efficient in extracting the structural information within an image, and PSNR does not consider the structural composition of the image (*Rouse & Hemami, 2008*). The structural similarity index metric (SSIM) was proposed by (*Wang et al., 2004*) to measure the structural similarity between images by comparing the contrast, luminance, and structural details within the reference image.

An image $I_r$ with total pixels $P$; the contrast $C_I$, and luminance $L_I$ can be denoted as the standard deviation and the mean of the image intensity given by:

$$L_I = \frac{1}{M} \sum_{i=1}^{M} I_r(i) \tag{8}$$

$$C_I = \sqrt{\left( \frac{1}{M-1} \sum_{i=1}^{M} \left( I_r(i) - L_I \right)^2 \right)} \tag{9}$$

The $i^{th}$ pixel of the reference image is denoted by $I_r(i)$. The comparisons based on the contrast and luminance between the reference image $I_r$ and the estimated image $\hat{I}$ are:

$$Com_l\left(I_r, \hat{I}\right) = \frac{2L_{I_r} L_{\hat{I}} + \mu_1}{L_{I_r}^2 + L_{\hat{I}}^2 + \mu_1} \tag{10}$$

$$Com_c\left(I_r, \hat{I}\right) = \frac{2C_{I_r} C_{\hat{I}} + \mu_2}{C_{I_r}^2 + C_{\hat{I}}^2 + \mu_2} \tag{11}$$

Where $\mu_1 = (k_1 S)^2$ and $\mu_2 = (k_2 S)^2$, these constant terms ensure stability by ensuring $k_1 << 1$ and $k_2 << 1$.

Normalized pixel values $I_r - L_{I_r}/C_{I_r}$ represent the image structure, while the inner product of these is the equivalent of structural similarity between the reference image $I_r$ and the estimated image $\hat{I}$. The covariance $\sigma_{I_r,)I}$ is given by:

$$\sigma_{I_r,\hat{I}} = \frac{1}{M-1} \sum_{i=1}^{M} \left(I_r(i) - L_{I_r}\right)\left(\hat{I}(i) - L_{\hat{I}}\right) \tag{12}$$

Function for structural comparison $Com_s\left(I_r, \hat{I}\right)$ is given by:

$$Com_s\left(I_r, \hat{I}\right) = \frac{\sigma_{I_r,\hat{I}} + \mu_3}{C_{I_r} C_{\hat{I}} + \mu_3} \tag{13}$$

Where $\mu_3$ is stability constant, the final structural similarity index (SSIM) is given by:

$$SSIM\left(I_r, \hat{I}\right) = \left\{Com_l\left(I_r, \hat{I}\right)\right\}^{\alpha}\left\{Com_c\left(I_r, \hat{I}\right)\right\}^{\beta}\left\{Com_s\left(I_r, \hat{I}\right)\right\}^{\gamma} \tag{14}$$

The control parameters $\alpha$, $\beta$ and $\gamma$ can be adjusted to increase the importance of luminance, contrast, and structural comparison in calculating the *SSIM*.

Conventionally, PSNR is used in computer vision tasks for evaluation, but SSIM is based on human perception of structural information within an image. Thus this method is widely used for comparing the structural similarity between images (*Blau & Michaeli, 2018*; *Sara, Akter & Uddin, 2019*). In medical images where the variance or luminance of the reference images are low, SSIM could be very unstable, thus reporting false results; however, this is not the case for natural images (*Pambrun & Noumeir, 2015*).

### Opinion scoring

Opinion scoring is a qualitative method, which lies in the subjective category of IQA. In this method, the quality testers are asked to grade the quality of images based on specific criteria, e.g., sharpness, natural look, and color, where the final graded score is the mean of the rated scores.

This method has limitations such as non-linearity between the scores, variation in results due to changes in test criteria, and human error. In SR, certain methods have reported good objective quality scores but scored poorly in subjective results, especially in human face reconstruction (*Ledig et al., 2017*; *Nasrollahi & Moeslund, 2014*; *Chen et al., 2018b*). Thus, the opinion scoring method is also used in studies (*Wei, Yuan & Cai, 1999*; *Deng, 2018*; *Ravì et al., 2018, 2019*; *Vasu, Thekke Madam & Rajagopalan, 2019*) to measure the quality of human perception.

### Perceptual quality

Opinion scoring used human raters for manual evaluation of the images; while this method can provide accurate results as far as human perception is concerned, this method requires many resources, especially large datasets (*Viswanathan & Viswanathan, 2005*). Initially (*Kim & Lee, 2017*) proposed a CNN-based full reference image quality assessment (FR-IQA) model where human behavior was learned using an IQA database that

contained distorted images, subjective scores, and error maps, and this method was called DeepQA.

In *Ma et al. (2017a)*, the authors used quality-discriminable image pairs (DIP) for training, and the system was called dipQA (DIP inferred quality index); they used RankNet with L2R algorithm to learn blind opinion IQA, whereas in *Ma et al. (2018)* a multi-task end-to-end optimized deep neural network (MEON) was proposed. MEON used two stages, in the first stage, distortion type learning using large datasets already available, and in the second stage, the output of the first stage was used to train the quality assessment network using stochastic gradient descent. In *Talebi & Milanfar (2018)*, the authors used CNN to develop a no-reference IQA method known as NIMA; NIMA was trained on pixel-level and aesthetic quality datasets.

RankIQA (*Liu, Van De Weijer & Bagdanov, 2017*) trained a Siamese network to grade the quality of images using datasets with known image distortions, CNNs were used to learn the IQA, and this method even outperformed full-reference methods without using the reference image. IQA proposed in *Bosse et al. (2018)* included ten convolution layers and five pooling layers for feature extraction while there were two fully connected layers for regression; this method performed significantly well for both no-reference and full-reference IQA.

Even though opinion scoring and perceptual quality-based methods do exhibit human perception in IQA, but the quality we require is still an open question (i.e., if we want images to be more natural or similar to the reference image); thus, PSNR and SSIM are the primarily used methods in computer vision and SR.

### Task-based evaluation

Although the primary purpose of image SR is to achieve better resolution, as mentioned earlier, SR is also helpful in other computer vision tasks (*Kim, Lee & Lee, 2016b*; *Tao et al., 2017*; *Teh et al., 2020*; *Liu et al., 2018a*). The performance achieved in these can indirectly measure the performance of the SR methods used in those tasks. In the case of medical images, the researchers used the original and SR constructed images to see the performance in the training and prediction phases. In general, computer vision tasks such as classification (*Krizhevsky, Sutskever & Hinton, 2012*; *Cai et al., 2019*), face recognition (*Nasrollahi & Moeslund, 2014*; *Liu et al., 2015*; *Chen et al., 2018b*), and object segmentation (*Martin et al., 2001*; *Lin et al., 2014*; *Wang et al., 2018b*) can be done using SR images. The performance of these computer vision tasks can be used as a metric to assess the performance of the SR method.

### Miscellaneous IQA methods

The development of IQA methods is an open field, and in recent years various researchers have proposed SR metrics, but these methods were not used widely by the SR community. Feature similarity (FSIM) index metric (*Zhang et al., 2011*) evaluates image quality by extracting feature points considered by the human visual system based on gradient magnitude and phase congruency. The multi-scale structural similarity (MS-SSIM) (*Wang, Simoncelli & Bovik, 2003*) used multi-scale to incorporate variations in the viewing

**Table 1 Comparison of image quality metrics for super-resolution.**

| Method | Strengths | Weaknesses |
|---|---|---|
| PSNR | • Most commonly used quality assessment metric; thus, it is easy to compare results with other methods. <br> • Quantitative and is based on MSE | • Since this metric is pixel-based, the overall score could be misleading in some cases where two images could be visually different, but the PSNR would still be high. <br> • This method does not give consider the structural information within the image |
| SSIM | • After PSNR, this metric is the most commonly used IQA metric; thus, comparing results with other methods is easier. <br> • Quantitatively scores an image based on its structural similarity with the original image with the possibility to change the weights of luminance, contrast, and structural comparison. | • SSIM is unstable in cases where the variance or luminance of the reference image is low; thus, in medical imaging, this metric could give inconsistent results. |
| Opinion Scoring | • Opinion scoring is a subjective quality metric, and human testers grade image quality based on predefined parameters such as sharpness, color, and natural look. <br> • This method is particularly suitable for human face reconstruction methods. | • Limitations include non-linear scoring among the testers, human error, and changes in test parameters. <br> • Scoring takes much time, especially for large datasets |
| Perceptual Quality | • This method is similar to opinion scoring, but human testers are replaced by models that learn the behavior of testers using deep learning. <br> • Very fast compared to opinion scoring. | • Requires additional resources for training the network to learn the features for the quality assessment network. <br> • It depends on annotated datasets to learn human behavior. |
| Task-based Evaluation | • This metric is appropriate if the SR images are used to perform another task, for example, object detection/classification and diagnosis. <br> • It helps in measuring the performance of the whole task, which uses SR images. | • Highly dependent on the performance of the associated task. <br> • Same SR images will give different scores if there is a change in task parameters. |

conditions to measure the image quality and proposed that MS-SSIM provides form flexibility in the measurement of image quality than single-scale SSIM. In *Li & Bovik (2010)*, the authors claimed that SSIM and MS-SSIM do not perform well on distorted and noisy images; thus, they used a four-component-based weighted method that adjusted the weight of scores based on the local feature, whereas in the case of contrast-distorted images *Yao & Liu (2018)* like TID2013 and CSIQ datasets SSIM does not perform well.

According to *Blau & Michaeli (2018)*, the perceptual quality and image distortion are at odds with each other; as the distortion decreases, the perceptual quality should also be worse; thus, the accurate measurement of SR image quality is still an open area of research.

The comparison of image quality assessment metrics for super-resolution is shown in Table 1. It depends on the requirements of the methods; most of the methods use PSNR and SSIM to evaluate the performance as these are quantitative methods.

## Operating color channels

In most datasets, RGB color space is used; thus, SR methods mostly employ RGB images, YCbCr space is also used in SR (*Dong et al., 2016*). The Y component in YCbCr is the luminance component, which represents the light intensity, while Cb and Cr are the

chrominance components (i.e., blue-differenced and red-differenced Chroma channels) (*Shaik et al., 2015*). In recent years, most of the SR challenges and datasets use the RGB color space, limiting the use of RGB space for comparison with state of the art. Furthermore, the results of IQA based on PSNR vary if the color space in the testing stage is different from the training/evaluation stage.

## Details of the reference dataset

The datasets used in evaluating the SR algorithms are summarized in this section; the various datasets discussed in this section vary in the total number of example images, image resolution, quality, and imaging hardware setup. A few of the datasets comprise paired LR-HR images for training and testing SR algorithms. In contrast, the rest of the datasets include HR images, and the corresponding LR images are usually generated by using bicubic interpolation with antialiasing as performed in *Shi et al. (2016)*, *Zhang & An (2017)* and *Shocher, Cohen & Irani (2018)*. Matlab function imresize (I, scale), where the default method is bicubic interpolation with antialiasing, and scale is the downsampling factor input to the function.

Table 2 comprises a list of datasets frequently used in SR and information on total image count, image format, pixel count, HR resolution, type of dataset, and classes of images.

Most of the datasets for SR are unpaired data, and the LR images are generated using various scale factors using bicubic interpolation with antialiasing. Other than the mentioned datasets in Table 2, datasets like General-100 (*Dong, Loy & Tang, 2016*), L20 (*Timofte, Rothe & Van Gool, 2016*) and ImageNet (*Deng et al., 2009*) are also used in computer vision tasks. In recent times, researchers have preferred the use of multiple datasets for training/evaluation and testing the SR models; for instance, in *Bashir & Ghouri (2014)*, *Lai et al. (2017)*, *Sajjadi, Scholkopf & Hirsch (2017)* and *Tong et al. (2017)*, the researchers used SET5, SET14, BSDS100 and URBAN100 for training and testing.

## Super-resolution challenges

The most prominent SR challenges NTIRE (*Agustsson & Timofte, 2017*; *Timofte et al., 2017*), and PIRM (*Blau et al., 2018*), are discussed in this section.

The New Trends in Image Restoration and Enhancement (NTIRE) challenge (*Agustsson & Timofte, 2017*; *Timofte et al., 2017*) was in collaboration with the Conference on Computer Vision and Pattern Recognition (CVPR). NTIRE includes various challenges like colorization, image denoising, and SR. In the case of SR, the DIV2K dataset (*Agustsson & Timofte, 2017*) was used, which included bicubic downscaled image pairs and blind images with realistic but unknown degradation. This dataset has been widely used to evaluate SR methods under known and unknown conditions to compare against the state-of-the-art methods.

The perceptual image restoration and manipulation (PIRM) challenges were in collaboration with the European Conference on Computer vision (ECCV), and like NTIRE, it contained multiple challenges. Apart from the three challenges mentioned in NTIRE, PIRM also focused on SR for smartphones and compared perceptual quality with generation accuracy (*Blau et al., 2018*). As mentioned by *Blau & Michaeli (2018)*, the

**Table 2  List of benchmark datasets used in super-resolution.**

| Name | Number of images/pairs | Image format | Type | Resolution | Details of images |
|---|---|---|---|---|---|
| BSD100 (*Martin et al., 2001*) | 100 | PNG | Unpaired | (480, 320) | 100 images of animals, people, buildings, scenic views etc. |
| BSDS300 (*Martin et al., 2001*) | 300 | JPG | Unpaired | (430, 370) | 300 images of animals, people, buildings, scenic views, plants, etc. |
| BSDS500 (*Arbeláez et al., 2010*) | 500 | JPG | Unpaired | (430, 370) | Extended version of BSD 300 with additional 200 images |
| CelebA (*Liu et al., 2015*) | 202,599 | PNG | Unpaired | (2048, 1024) | Over 40 attribute defined categories of celebrities |
| DIV2K (*Agustsson & Timofte, 2017*) | 1,000 | PNG | Paired | (2048, 1024) | Objects, People, Animals, scenery, nature |
| Manga109 (*Fujimoto et al., 2016*) | 109 | PNG | Unpaired | (800, 1150) | 109 manga volumes drawn by professional manga artists in Japan |
| MS-COCO (*Lin et al., 2014*) | 164,000 | JPG | Unpaired | (640, 480) | Labeled objects with over 80 object categories |
| OutdoorScene (*Wang et al., 2018b*) | 10,624 | PNG | Unpaired | (550, 450) | Outdoor scenes including plants, animals, sceneries, water reservoirs, etc. |
| PIRM (*Blau et al., 2018*) | 200 | PNG | Unpaired | (600, 500) | Sceneries, people, flowers, etc. |
| Set14 (*Zeyde, Elad & Protter, 2012*) | 14 | PNG | Unpaired | (500, 450) | Faces, animals, flowers, animated characters, insects, etc. |
| Set5 (*Bevilacqua et al., 2012*) | 5 | PNG | Unpaired | (300, 340) | Only 5 images including, butterfly, baby, bird, head, and women. |
| T91 (*Yang et al., 2010*) | 91 | PNG | Unpaired | (250, 200) | 91 images of fruits, cars, faces, etc. |
| Urban100 (*Huang, Singh & Ahuja, 2015*) | 100 | PNG | Unpaired | (1000, 800) | Urban buildings, architecture |
| VOC2012 (*Everingham et al., 2014*) | 11,530 | JPG | Unpaired | (500, 400) | Labelled objects with over 20 classes |

models that focus on distortion often give visually unpleasant SR images, while the models focusing on the perceptual image quality do not perform well on information fidelity. Using the image quality metrics NIQE (*Mittal, Soundararajan & Bovik, 2013*) and (*Ma et al., 2017b*), the methods that performed best in achieving perceptual quality (*Blau & Michaeli, 2018*) was the winner. In contrast, in a sub-challenge (*Ignatov et al., 2018b*), SR methods were evaluated using limited resources to evaluate SR performance for smartphones using the PSNR, MS-SSIM, and opinion scoring metrics. Thus, PIRM encouraged the researchers to explore the perception-distortion tradeoff domain and SR for smartphones.

## SURVEY METHODOLOGY

The majority of the studies included in this review paper are peer-reviewed publications to ensure the validity of the methods; these studies include conference proceedings and journal papers. The included papers include early access and a published version of recent papers for super-resolution from 2008 to 2021. However, some in classical methods, some papers, and initial papers on image SR were included before this range to develop the review and give the background of the classical methods developed before the deep learning-based methods overtook the field. Google Scholar, IEEE Xplore, and Science Direct were queried to collect the initial list of papers in this research. Specific keywords

**Table 3 Inclusion and exclusion criteria.**

| Section | Inclusion | Exclusion |
|---|---|---|
| Introduction | Methods that defined image interpolation and performed some practical form of image interpolation, i.e., super-resolution | • Studies that solely defined model<br>• Review articles |
| Classical Methods | Methods that performed pixel, neighborhood, or any classical image interpolation | • Application research where applications of classical methods were discussed<br>• Review papers |
| Deep learning-based methods | Development of image super-resolution using deep learning methods, including review papers | • Papers that emphasize video super-resolution as these papers give priority to frame per second (FPS) and inference time were not included |
| Applications | Direct applications of super-resolution methods in the six fields defined in "Domain-Specific Applications of Super-Resolution" were included | • Applications that combined other methods with image super-resolution and SR was a limited part were not included<br>• Review papers |

were used to search the databases and based on the abstract. A further selection of papers was made using the reference sections of the selected papers as they contain additional relevant studies in image super-resolution. The last query was made on May 08, 2021. The collected papers were segregated based on their relevance with the Section; for example, papers with supervised learning were stored separately for review in "Supervised Super-Resolution", and studies highlighting the applications of SR methods were grouped for discussion in "Domain-Specific Applications of Super-Resolution".

The relevant search terms include image super-resolution, super-resolution, deep learning super-resolution, convolutional neural networks, image upsampling methods, super-resolution frameworks, supervised super-resolution, unsupervised super-resolution, super-resolution review, image interpolation, pixel-based methods, super-resolution application, assisted diagnosis using deep learning. The search keywords were not limited to single-image SR because our target was to report other aspects of super-resolution, including classical methods, applications, and datasets for SR.

Logical operators and wildcards were used to combine the keywords further and perform the additional search. Initial screening of the collected papers was performed following the inclusion/exclusion criteria shown in Table 3. The whole process is graphically shown in Fig. 3, where 653 studies were collected over one year. A total of 242 studies were included from the initial 653 collected research studies.

## CONVENTIONAL METHODS OF SUPER-RESOLUTION

Classical methods of SR are briefly discussed in this section to encompass the overall development cycle of the SR. The classical methods include prediction-based, edge-based, statistical, patch-based and sparse representation methods.

The primary methods were based on prediction, and the first method (*Duchon, 1979*) was based on Lanczos filtering, which filtered the digital data using sigma factors (with

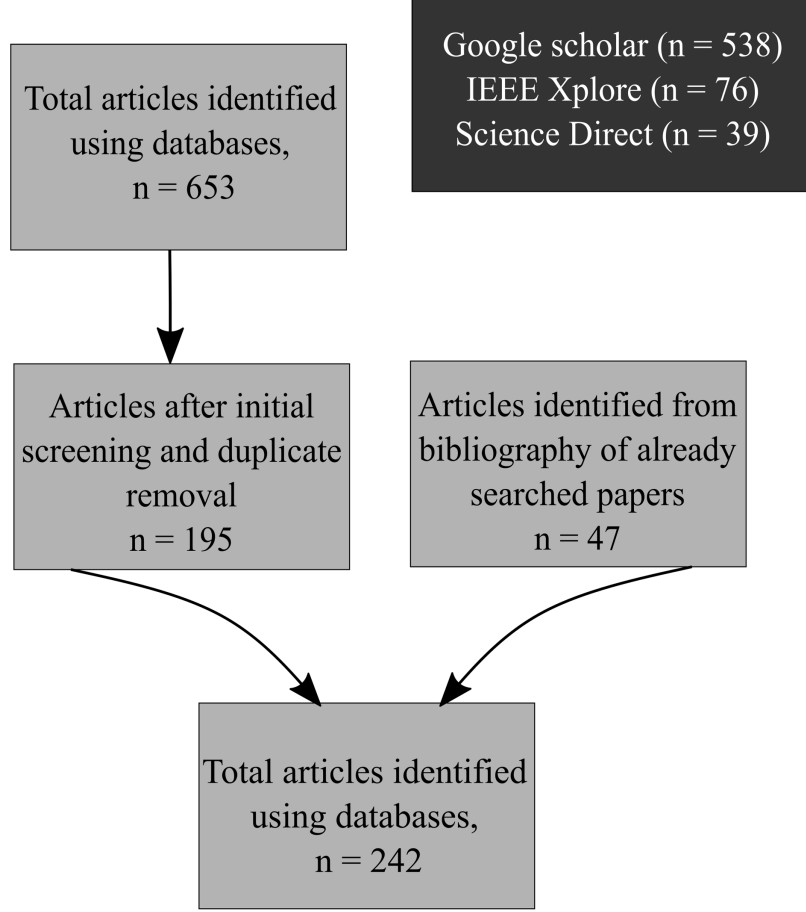

**Figure 3 Methodology for the collection of studies.** Sample details based on inclusion/exclusion criteria defined in Table 5.

modifiable weight function), and a similar frequency-domain filtering approach was used in *Tsai & Huang (1984)* for image resampling. In contrast, cubic convolution (*Keys, 1981*) was used for resampling the image data, and the results showed that this prediction method was more accurate than the nearest-neighbor prediction algorithm and linear interpolation of image data (*Parker, Kenyon & Troxel, 1983*). In *Tsai & Huang (1984)*, the authors did not consider the blur in the imaging process, while (*Irani & Peleg, 1991*) used the knowledge of the imaging process and the relative displacements for image interpolation when the sampling rate was kept constant and this method reduced to deblurring.

The patch-based approach was used in *Freeman, Jones & Pasztor (2002)*; the authors used a training set where various patches within the training set were extracted as training patterns, which helped generate detailed high-frequency images using the patch texture information. In *Chang, Yeung & Xiong (2004)*, the authors used locally linear embedding to use local patches for generating high-resolution images based on the local patch features. In contrast, (*Glasner, Bagon & Irani, 2009*) used the concept of reoccurrence of geometrically similar patches in natural images to select the best possible pixel value based

on the patch redundancy on the same scales. In *Baker & Kanade (2002)*, the authors introduced the concept of hallucination, where they extracted local features within the LR image first and used these to map the HR image.

Edge-based methods use edge smoothness priors to upsample images, and in *Sun, Xu & Shum (2008)*, a generic image prior, gradient prior profile was used to smoothen the edges within an image to achieve super-resolution in natural images. In *Freedman & Fattal (2011)*, the authors used specially designed filters to search for similar patches using the local self-similarity observation, which performed lower nearest patch computations; this method was able to reconstruct realistic-looking edges, whereas it performed poorly in clustered regions with fine details.

Statistical methods were used to perform image super-resolution (*Kim & Kwon, 2010*), where the authors used Kernel ridge regression (KRR) with gradient descent to learn the mapping function from the image example pairs. Adaptive regularization was used to supervise the energy change during the image resampling iterative process. This provided more accurate results as the energy map was used to limit the energy change per iteration, which reduced the noise while maintaining the perceptual quality (*Xiong, Sun & Wu, 2010*) while *Yang et al. (2010)* and *Yang et al. (2008)* used sparse representation methods to perform image super-resolution which used the concept of compressed sensing.

The robust SR method proposed in *Zomet, Rav-Acha & Peleg (2001)* used the information of outliers to improve the performance of SR in patches where other methods introduce noise due to these outliers. Additionally, *Yang et al. (2007)* proposed a post-processing model that enhanced the resolution of a set of images using a single reference image up to 100x scaling factor. Another way to achieve SR is to use LR images to achieve a single HR image (*Tipping & Bishop, 2003*). The conventional upsampling methods, such as interpolation-based, use the information within the LR image to generate HR images, and these methods do not add any new information to the image (*Farsiu et al., 2004a*). Furthermore, they also introduce some inherent problems, such as noise amplification and blur enhancement. Thus, in recent years, the researchers have shifted to learning-based upsampling methods explored in "Supervised Super-Resolution".

## SUPERVISED SUPER-RESOLUTION

Various deep learning methods were developed over the years to solve the SR problem; in this section, the models discussed are trained using both low and high-resolution images (LR–HR pairs). Although there are significant differences in the supervised SR models, and the models can be classified based on the components like the upsampling method employed, deep learning network, learning algorithm, and model frameworks. Any supervised image SR model is based on the combinations of these components, and in this section, we summarize the employed methods for these four components in light of recent supervised image SR research studies.

The component-based review of various methods is performed in this section, and the basic overview of the models is shown in Fig. 1.

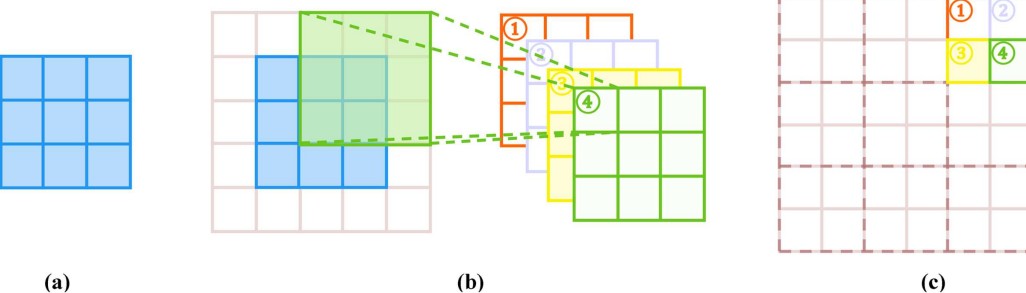

(a)                                        (b)                                        (c)

**Figure 4 Sub-pixel layer.** *Blue* **color represents the input convolution, and output feature maps are represented in other colors.** (A) Input. (B) Convolution. (C) Reshaping.

## Upsampling methods

The upsampling is essential in deep learning-based SR methods such as its positioning, and the method performed for upsampling has a significant impact on the training and test performance of the model. There are some commonly used methods (*Yang et al., 2010, 2008*; *Lee, Yang & Oh, 2015*; *Timofte, De Smet & Van Gool, 2015*), which use the conventional CNNs for end-to-end learning. In this subsection, various deep learning-based upsampling layers are discussed.

As mentioned in "Conventional Methods of Super-Resolution", the interpolation-based methods of upsampling do not add any new information; hence, learning-based methods are used in image SR in the last decade.

### Sub-pixel layer

The end-to-end learning layer (*Shi et al., 2016*), called the sub-pixel layer, performs upsampling by generating several additional channels using convolution, and by reshaping these channels, this layer performs upsampling, as shown in Fig. 4. In this layer, convolution is applied to $s_f^2$ where $s_f$ is the scaling factor, as shown in Fig. 4B. Since the input image size is $h \times w \times c$ where $h$ is height, $w$ is width, and $c$ depicts color channels, the resulting convolution is $h \times w \times cs_f^2$. In order to achieve the final image, a reshuffling (*Shi et al., 2016*) operation is performed to get the final output image $s_f h \times s_f w \times c$, as shown in Fig. 4C. Since it is an end-to-end layer, this layer is frequently used in SR models (*Ledig et al., 2017*; *Zhang, Zuo & Zhang, 2018*; *Ahn, Kang & Sohn, 2018a*; *Zhang et al., 2018b*).

This layer has a wide receptive field, which helps learn more contextual information that generates realistic details, whereas this layer may generate some false artifacts at the boundaries of complex patterns due to its uneven distribution of the respective field. Furthermore, predicting the neighborhood pixels in a block-type region sometimes results in unsmooth outputs that do not look realistic when compared with the true HR image; to address this issue, PixelTCL (*Gao et al., 2020*) was proposed that used the interdependent prediction layer, which used the information of the interlinked pixels during upsampling. The results were smooth and more realistic when compared with the ground truth image.

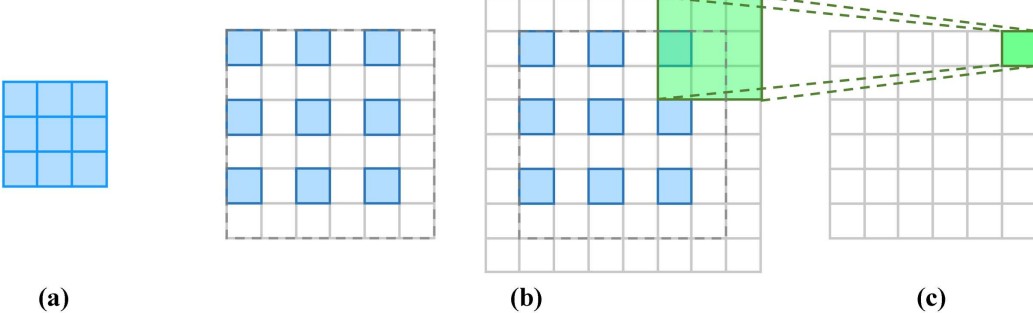

**(a)**                    **(b)**                    **(c)**

**Figure 5 Deconvolution layer. The *blue* color represents the input, and the *green* color represents the convolution operation.** (A) Input. (B) Expansion. (C) Convolution.

### Deconvolution layer

The deconvolution layer also referred to as transposed convolution layer (*Zeiler et al., 2010*), is the converse of the convolution, i.e., predicting the probable input HR-image based on the feature maps from the LR image. In this process, additional zeros are inserted to increase the resolution, and afterwards, convolution is performed. For instance, taking scaling factor 2 for the SR image, a convolution kernel of $3 \times 3$ (as shown in Figs. 5A, 5B and 5C), the input LR image is expanded twice by inserting zeros, convolution with the kernel is performed by using a stride and padding of 1.

The deconvolution layer is widely used in SR methods (*Sroubek, Cristobal & Flusser, 2008*; *Hugelier et al., 2016*; *Lam et al., 2017*; *Tong et al., 2017*; *Haris, Shakhnarovich & Ukita, 2018*), as it generates HR images in an end-to-end way, and it has compatibility with the vanilla convolution. As per *Odena, Dumoulin & Olah (2017)*, in some cases, this layer may cause the problem of uneven overlapping within the generated HR image as the patterns are replicated in a check-like format and may result in a non-realistic HR image, thereby decreasing the performance of the SR method.

### Meta upscaling

The scaling factor was predefined in the previously mentioned methods, thereby training multiple upsampling modules with different factors, which is often inefficient and is not the actual requirement of an SR method. A meta upscaling module (*Hu et al., 2019*) was proposed; this module uses arbitrary scaling factors to generate SR image-based in meta-learning. Meta scaling module projects every position in the required HR image to a small patch in the given LR feature maps $j \times j \times c_i$, where j is arbitrary, and ci is the total number of channels within the extracted feature map (in *Hu et al., 2019* this was 64). Additionally, it also generates the convolution weights $(j \times j \times (c_i \times c_o))$, where $c_o$ represents the output image channels, and it is usually 3. Thus, the meta upscaling module continuously uses arbitrary scaling factors within a single model and using a substantial training set, a large number of factors are simultaneously trained. The performance of this layer even surpasses the results produced with fixed factor models, and even though this module predicts the weights during the inference time, the overall execution time for weight prediction is 100 times less than the total time required for feature extraction

**Table 4  Comparison of upsampling methods.**

| Method | Strengths | Weaknesses |
|---|---|---|
| Sub-pixel layer | • It uses convolution in an end-to-end manner, so it is frequently used in SR models.<br>• This layer has a wide receptive field, which helps learn more contextual information | • This layer may generate some false artifacts at the boundaries of complex patterns due to its uneven distribution of the respective field.<br>• Upscaling factor is fixed |
| Deconvolution layer | • This layer is most commonly used in SR methods, and it generates HR images in an end-to-end manner.<br>• Compatible with vanilla convolution | • In some cases, due to uneven overlapping within the generated HR image, the patterns are replicated in a check-like format and may result in a non-realistic HR image.<br>• Upscaling factor is fixed |
| Meta upscaling | • This method uses arbitrary scaling factors to generate the SR image.<br>• Extracts more information from the LR feature maps, which helps to construct an HR image using meta upscaling in the last layer of the SR models, which makes this method an end-to-end SR approach | • The whole process may become unstable for high-scale factors as it predicts the convolution weights for every single pixel independent of the image information within those pixels. |

(*Hu et al., 2019*). In cases where there is a need for larger magnifications, this module may become unstable as it predicts the convolution weights for every pixel independent of the image information within those pixels.

This upscaling method is frequently used in recent years, particularly in post-upsampling frameworks ("Post-Upsampling SR"). The high-level representations extracted from the low-level information are used to construct an HR image using meta upscaling in the last layer of the model, making this method an end-to-end SR approach.

The comparison of upsampling methods is shown in Table 4; most SR methods use deconvolution or sub-pixel layers for upscaling. However, for multiple scale factors, meta upscaling is used.

## Deep learning SR networks

The network design and advancements in design architecture are recent trends in deep learning, and in SR, researchers have tried several design implications along with the SR framework (as seen in "SR Frameworks") for designing the overall SR network. Some of the fundamental and recent network designs are discussed in this section.

### *Recursive learning*

One of the basic network-based learning strategies is to use the same module for recursively learning high-level features. This method also minimizes the parameters as the strategy is based on the same module being updated recursively, as shown in Fig. 6A.

One of the most used recursive networks is the Deeply-recursive Convolutional Network (DRCN) (*Kim, Lee & Lee, 2016b*). Utilizing a single convolution layer DRCN reaches up to a $41 \times 41$ repetitive field without requiring additional parameters, which is very deep compared to the Super-resolution Convolution Neural Network SRCNN (*Thapa et al., 2016*) ($13 \times 13$). The Deep Recursive Residual Network (DRRN)

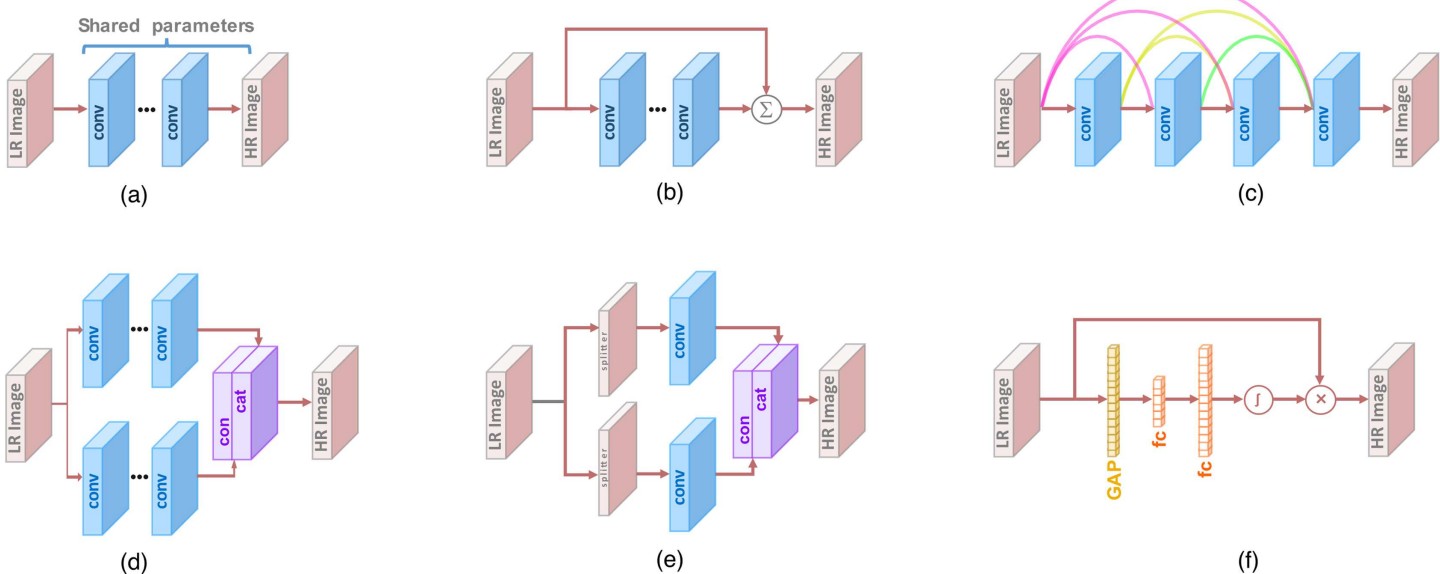

**Figure 6 Deep-learning network structures for super-resolution.** (A) Recursive learning, (B) residual learning, (C) dense connection-based learning, (D) multiscale learning, (E) advanced convolution-based learning, (F) attention-based learning.

(*Tai, Yang & Liu, 2017*) utilized a ResBlock (*He et al., 2016*) as part of the recursive module for a total of 25 recursions and was reported to achieve better performance than the baseline ResBlock. Using the concept of DRCN, *Tai et al. (2017)* proposed a memory block-based method MemNet which contained six recursive ResBlocks, whereas the Cascading Residual Network (CARN) (*Ahn, Kang & Sohn, 2018a*) also used ResBlocks as recursive units. In this approach, the network shares the weights globally in recursion using an iterative up-and-down sampling-based approach. Apart from end-to-end recursions, the researchers also used Dual-state Recurrent Network (DSRN) (*Han et al., 2018*), which shared the signals between the LR and generated HR states within the network.

Overall, while reducing the parameters, recursive learning networks can learn the complex representation of the data at the cost of computational performance. Additionally, the increase in computational requirements may result in an exploding or vanishing gradient. Thus, recursive learning is often used in combination with multi-supervision or residual learning for minimizing the risk of exploding or vanishing gradient (*Kim, Lee & Lee, 2016b*; *Tai et al., 2017*; *Tai, Yang & Liu, 2017*; *Han et al., 2018*).

### Residual learning

Residual learning was widely used in the field of SR (*Bevilacqua et al., 2012*; *Timofte, De Smet & Van Gool, 2015*; *Timofte, De & Van Gool, 2013*), until ResNet (*He et al., 2016*) was proposed for learning residuals, as shown in Fig. 6B. Overall, there are two approaches, local and global residual learning.

The local residual learning approach mitigates the degradation problem (*He et al., 2016*) caused by increased network depth. Furthermore, the local residual learning also improved the learning rate and reduced the training difficulty; this is frequently used in the SR field (*Protter et al., 2009*; *Mao, Shen & Bin, 2016*; *Han et al., 2018*; *Li et al., 2018*).

The global residual learning is an approach used in which the input and the final output are correlated, and in image SR, the output HR is highly correlated with the input LR image; thus, learning the global residuals between LR and HR image is significant in SR. In global residual learning, the model only learns the residual map that transforms the LR image into an HR image by generating the missing high-frequency details in the LR image. Furthermore, the residuals are minimal; thereby, the learning difficulty and model complexity are significantly reduced in global residual-based learning. This method is also frequently used in SR methods (*Kim, Lee & Lee, 2016a*; *Tai, Yang & Liu, 2017*; *Tai et al., 2017*; *Hui, Wang & Gao, 2018*).

Overall, both methods use residuals to connect the input image with the output HR image; in the case of global residual learning, the connection is directly made, which in local residual learning various layers of different depth to connect the input (using local residuals) with the output.

### Dense connection-based learning

This learning method uses dense blocks to address SR, like DenseNet (*Huang et al., 2017*). The dense block utilizes all the features maps generated by the previous layers as inputs and its feature inputs, leading to $l(l-1)/2$ connections in an l-layer $(l \geq 2)$ dense block. Using dense blocks will increase the reusability of the features while resolving the gradient vanishing problem. Furthermore, the dense connections also minimize the model size by utilizing a small growth rate and enfolding the channels using concatenated input features.

Dense connections are used in SR to connect the low-level and high-level features maps for reconstructing a high-quality fine-detailed HR image, as shown in Fig. 6C. SRDenseNet (*Tong et al., 2017*) proposed a 69-layer network containing dense connections within the dense blocks and dense connections among the dense blocks. In SRDenseNet, the feature maps from the prior blocks and the feature maps were used as inputs of all preceding blocks. RDN (*Zhang et al., 2018b*), CARN (*Ahn, Kang & Sohn, 2018a*), MemNet (*Tai et al., 2017*) and ESRGAN (*Wang et al., 2019c*) also used layer or block-level dense connection, while DBPN (*Wang et al., 2018b*) only used the dense connection between the upsampling and downsampling units.

### Multi-path learning

In multi-path learning, the features are transferred to multiple paths for different representations, and these representations are later combined to gain improved performance. Scale-specific, local, and global multi-path learnings are the main types.

For different scales, the super-resolution models use different feature extraction; in *Lim et al. (2017)*, the authors proposed a single network-based multi-path learning for

multiple scales. The intermediate layers of the model were shared for feature extraction, while scale-specific paths, including pre-processing and upsampling, were at the end of the models, i.e., the start and end of the network. During training, the scale relative paths are enabled and updated accordingly, and the proposed deep super-resolution MDSR method (*Lim et al., 2017*) also decreases the overall model size because of the sharing of parameters across the scales. Like MDSR, a similar multi-path-based approach is also implemented in ProSR and CARN.

Local multi-path learning is inspired using a new block, the inception module (*Szegedy et al., 2015*), for multi-scale feature extraction, as performed in MSRN (*Li et al., 2018*) (shown in Fig. 6D). The additional block consists of $3 \times 3$ and $5 \times 5$ kernel size convolution layers, which simultaneously extracts the features. After combining the outputs of the two convolution layers, the final output goes through a $5 \times 5$ kernel convolution. Furthermore, a path links the input and output by element-wise addition and uses this local multi-path learning; this method extracts features efficiently than multi-scale learning.

Another variation of multi-path learning is global multi-path learning; in this method, various features are extracted from multi-paths that can interact. In DSRN (*Han et al., 2018*), there are two paths for extracting low and high-level information, and there is a continuous sharing of features for improved learning. In contrast, in pixel recursive SR (*Dahl, Norouzi & Shlens, 2017*), a conditioning path is responsible for extracting global structures, and the prior path further finds the serial codependence among the generated pixels. A different method was employed by *Ren, El-Khamy & Lee (2017)*, where multi-path learning was performed for unbalanced structures, which were later combined in the final layer to get the SR output.

### Advanced convolution-based learning

In SR, the methods explored depend on the convolution operation, and various research studies have attempted to modify the convolution operation for better performance. In recent years, research studies have shown that group convolution, as shown in Fig. 6E, decreased the total number of parameters at the cost of small loops in performance (*Hui, Wang & Gao, 2018*; *Johnson, Alahi & Li, 2016*). In CARN-M (*Ahn, Kang & Sohn, 2018a*) and IDN (*Hui, Wang & Gao, 2018*), group convolution was used instead of vanilla convolution. In dilated convolution, the contextual information is used to generate realistic-looking SR images (*Zhang et al., 2017*); dilated convolution was used to double the receptive field, resulting in better results.

Another type of convolution is depthwise separable convolution (*Howard et al., 2009*); although this convolution significantly reduces the total number of parameters, it reduces the overall performance.

### Attention-based learning

In deep learning, attention learning is the idea where certain factors are given more preference, which processes the data than others; here, two types of attention-based learning mechanisms are discussed in SR. In channel attention, a particular block is added

in the model where global average pooling (GAP) squeezes the input channels; two fully connected layers process these constants to generate channel-wise residuals (*Hu, Shen & Sun, 2018*), as shown in Fig. 6F. This technique has been incorporated in SR, known as RCAN (*Zhang et al., 2018a*), which has improved performance. Instead of GAP, *Dai et al. (2019)* used the second-order channel attention (SPCA) module, which used second-order feature metric for extracting more data representation using channel-based attention

In SR, most of the models use local fields for the generation of SR pixels, while in a few cases, some textures or patches which are far apart are necessary for generating accurate local patches. In *Zhang et al. (2019b)*, local and non-local attention blocks were used to extract local and non-local representations between pixel data. Similarly, the non-local attention technique was incorporated by *Dai et al. (2019)* to capture contextual information using a non-local attention method. Chen et al. proposed an SR reconstruction method with feature maps to facilitate the reconstruction of the image using an attention mechanism (*Chen et al., 2021*), while Yang et al. proposed a channel attention and spatial graph convolutional network (CASGCN) for a more robust feature obtaining and feature correlations modeling (*Yang & Qi, 2021*).

### Wavelet transform-based learning

Wavelet transform (WT) (*Daubechies & Bates, 1993*; *Griffel & Daubechies, 1995*) represents textures using high-frequency sub-bands and global structural information in low-frequency sub-bands in a highly efficient way. WT was used in SR to generate the residuals of the HR sub-bands using the sub-bands of the interpolated LR wavelet. Using the WT, the LR image is decomposed, while the inverse WT provides the reconstruction of the HR image in SR. Other examples of WT based SR are Wavelet-based residual attention network (WRAN) (*Xue et al., 2020*), multi-level wavelet CNN (MWCNN) (*Liu et al., 2018b*) and (*Ma et al., 2019*); these approaches used a hybrid approach by combining WT with other learning methods to improve the overall performance.

### Region-recursive-based learning

In SR, most methods follow the underlying assumption that it is a pixel-independent process; thus, there is no priority to the interdependence among the generated pixels. Using the concept of PixelCNN (*Van Den Oord et al., 2016*; *Dahl, Norouzi & Shlens, 2017*) proposed a method for pixel recursive learning, which performed SR by pixel-by-pixel generation using two networks. The two networks (*Dahl, Norouzi & Shlens, 2017*) captured information about pixel dependence and global contextual information within the pixel recursive SR method. Using the mean opinion scoring-based evaluation method (*Dahl, Norouzi & Shlens, 2017*) performed well compared to other methods for generating SR face images using the pixel recursive method. The attention-based face hallucination method (*Cao et al., 2017a*) also utilized the concept of a path-based attention shifting mechanism to enhance the details in the local patches.

While the region-recursive methods perform marginally better than other methods, the recursive process exponentially increases the training difficulty and computation costs due to long propagation paths.

### Other methods

Other SR networks are also used by researchers, such as Desubpixel-based learning (*Vu et al., 2019*), xUnit-based learning (*Kligvasser, Shaham & Michaeli, 2018*) and Pyramid Pooling-based learning (*Zhao et al., 2017*).

To improve the computational speed, the desubpixel-based approach was used to extract features in a low-dimensional space, which does the inverse task of the sub-pixel layer. By segmenting the images spatially and using them as separate channels, the desubpixel-based learning avoids any information loss; after learning the data representations in low-dimensional space, the images are upsampled to get a high-resolution image. This technique is particularly efficient in applications with limited resources such as smartphones.

In xUnit learning, a spatial activation function was proposed for learning complicated features and textures. In xUnit, the ReLU operation was replaced by xUnit to generate the weight maps through Gaussian gating and convolution. The model size was decreased by 50% using xUnit at the cost of increased computational demand without compromising the SR performance (*Kligvasser, Shaham & Michaeli, 2018*).

## Learning strategies

Learning strategies also dictate the overall performance of any SR algorithm as the evaluations are dependent upon the choice of the learning strategy selected. In this section, recent research studies are discussed using the learning strategy utilized in SR, and some of the critical strategies are discussed in detail.

### Loss functions

For any application in deep learning, the selection of the loss functions is critical, and in SR, these functions are used to measure the error in the reconstruction of HR, which further helps optimize the model iteratively. Since the necessary element of the images is a pixel, initial research studies employed the pixel loss, L2, but it was evaluated that the pixel loss cannot wholly represent the quality of reconstruction (*Ghodrati et al., 2019*). Thus, in SR, different loss functions such as content loss (*Johnson, Alahi & Li, 2016*) or adversarial loss (*Ledig et al., 2017*) are used to measure the error in the generation these loss functions have been widely used in the field of SR. Various loss functions are explored in this section, and the notation follows the previously defined variables except where defined otherwise.

Content Loss. The perceptual quality, as mentioned previously, is essential in the evaluation of an SR model, and this loss was used in SR (*Johnson, Alahi & Li, 2016*; *Dosovitskiy & Brox, 2016*) to measure the differences between the generated and ground-truth images using an image classification network (N). Let the high-level data representation on the $l^{th}$ lth layer is $r^l(I)$, the content loss is defined as the Euclidean among the high-level representations of the two images $I$ and $\hat{I}$, where $I$ is the original image and $\hat{I}$ is the generated SR image as below:

$$\mathcal{L}\left(I, \hat{I}; N_c, l\right) = \frac{1}{h_l w_l c_l} \sqrt{\sum_{i,j,k} \left(r^l_{i,j,k}(\hat{I}) - r^l_{i,j,k}(I)\right)^2} \qquad (15)$$

Where $h_l$, $w_l$ and $c_l$ respectively are height, width, and several channels of the image representations in the $l$ layer.

Content loss aims to share information about image features from the image classification network $N_c$ to the SR network. This loss function ensures the visual similarity between the original image ($I$) and the generated image ($\hat{I}$) by comparing the content and not the individual pixels. Thus, this loss function helps in producing visually perceptible and more realistic looking images in the field of SR as in *Ledig et al. (2017)*, *Wang et al. (2018b)*, *Sajjadi, Scholkopf & Hirsch (2017)*, *Wang et al. (2019c)*, *Johnson, Alahi & Li (2016)* and *Bulat & Tzimiropoulos (2018)* where the networks used as pre-trained CNNs were ResNet (*He et al., 2016*) and VGG (*Simonyan & Zisserman, 2015*).

Adversarial Loss. In recent years, after the development of GANs (*Goodfellow et al., 2014*), GANs have received more consideration due to their ability to learn and self-supervise. A GAN combines dual networks performing generation and discrimination tasks, i.e., generating the actual output and using a discriminator network to evaluate the results of the generative network. While training the GANs, two continuous updates were performed, i.e. (i) Adjust the generator for better results while training the discriminator to discriminate more efficiently and (ii) Adjust the discriminator while training the generator. This is a recursive training network, and through many iterations of training and evaluation, the generator can generate the output that conforms to the distribution of the actual data. The discriminator is unable to differentiate between real and generated information.

In terms of image SR, the purpose of a generative network is to generate an HR image, while another discriminator network will be used to evaluate if the image is of the same distribution as the input data. This method was first introduced in SR as SRGAN (*Ledig et al., 2017*), the adversarial loss in *Ledig et al. (2017)* was represented by:

$$\mathcal{L}_{GAN\_CE\_g}\left(\hat{I}; D\right) = -\log D\left(\hat{I}\right) \qquad (16)$$

$$\mathcal{L}_{GAN\_CE\_d}\left(\hat{I}, I_s; D\right) = -\left\{\log D(I_s) + \log\left(1 - D\left(\hat{I}\right)\right)\right\} \qquad (17)$$

Where $\mathcal{L}_{GAN\_CE\_g}$. is the adversarial loss function of the generator in the SR model, while $\mathcal{L}_{GAN\_CE\_d}$ is the adversarial loss function of the discriminator $D$, which is a binary classifier. In (17), the randomly sampled ground truth image is denoted by $I_s$. The same loss functions were reported by *Sajjadi, Scholkopf & Hirsch (2017)*.

Other than binary classification error, the studies *Yuan et al. (2018)* and *Wang et al. (2018a)* used mean square error for improved training and better results compared to (*Ledig et al., 2017*), the loss functions are given in (18) and (19):

$$\mathcal{L}_{GAN\_LS\_g}\left(\hat{I}; D\right) = \left(D\left(\hat{I}\right) - 1\right)^2 \qquad (18)$$

$$\mathcal{L}_{GAN\_LS\_d}\left(\hat{I}, I_s; D\right) = \left\{ \left(D(\hat{I})\right)^2 + \left(D(I_s) - 1\right)^2 \right\} \tag{19}$$

Contrary to the loss functions mentioned in (18) and (19), (*Park et al., 2018*) showed that in some cases, pixel-level discriminator network generates high-frequency noise; thus, we used another discriminator network to evaluate the first discriminator network for high-frequency representations. Using the two discriminator networks, (*Park et al., 2018*) were able to capture all attributes accurately.

Various opinion scoring systems have been used regressively to test the performance of the SR model that uses adversarial loss. Although the SR models attained lower PSNR than the pixel-loss-based SR on perceptual quality metrics like opinion scoring, these adversarial loss-based SR methods scored very high (*Ledig et al., 2017*; *Sajjadi, Scholkopf & Hirsch, 2017*). The use of a discriminator as the control network for the generator GANs was able to regenerate some intricate patterns that were very difficult to learn using ordinary deep learning methods. The only drawback of the GANs is their training stability (*Arjovsky, Chintala & Bottou, 2017*; *Gulrajani et al., 2017*; *Lee et al., 2018a*; *Miyato et al., 2018*).

Pixel Loss. As evident from the name, this loss function performs a pixel-wise comparison between the reference image and the generated image, and there are two types of comparisons, i.e., an *L1* loss, which is also termed as mean absolute error and *L2* loss, which is the mean square error (*MSE*)

$$\mathcal{L}_{PIX\_L1}\left(I, \hat{I}\right) = \frac{1}{hwc} \sum_{i,j,k} \left| I_{i,j,k} - \hat{I}_{i,j,k} \right| \tag{20}$$

$$\mathcal{L}_{PIX\_L2}\left(I, \hat{I}\right) = \frac{1}{hwc} \sum_{i,j,k} \left| I_{i,j,k} - \hat{I}_{i,j,k} \right|^2 \tag{21}$$

The *L1* loss in some cases becomes numerically unstable to compute; thus, another variant of the L1 loss called the Charbonnier loss (*Farsiu et al., 2004b*; *Barron, 2017*, *2019*; *Lai et al., 2017*) is given by:

$$\mathcal{L}_{PIX\_CH}\left(I, \hat{I}\right) = \frac{1}{hwc} \sum_{i,j,k} \sqrt{\left| I_{i,j,k} - \hat{I}_{i,j,k} \right|^2 - e^2} \tag{22}$$

Here *e* is a constant which ensures numerical stability.

The pixel loss function ensures that the generated HR image $\hat{I}$ has the same pixel values as the HR image *I*. Furthermore, the L2 loss used the square of pixel-value errors, giving more weightage to high-value differences than lower ones; thus, this loss function may give either the too variable result (in case of outliers) or give too smooth results (in case of minimal error values). Therefore, the L1 loss function is widely used over L2 loss (*Zhao et al., 2016*; *Lim et al., 2017*; *Ahn, Kang & Sohn, 2018a*). Furthermore, the PSNR equation is closely related to the definition of L1 loss, and minimizing L1 loss always leads to increased PSNR. Thus, researchers have often used the L1 loss to maximize the PSNR; as

mentioned earlier, the pixel loss function does not cater to perceptual quality or textures. Thus, SR networks based on this loss function may have less high-frequency details, resulting in smooth but unrealistic HR images (*Wang, Simoncelli & Bovik, 2003*; *Wang et al., 2004*).

Style Reconstruction Loss. Ideally, the reconstructed HR image should have comparable styles to the actual HR image (colors, textures, gradient, contrast), thus using the research studies (*Sajjadi, Scholkopf & Hirsch, 2017*; *Gatys, Ecker & Bethge, 2015*), style reconstruction loss was used in SR to match the texture details of the reference image with the generated image. The correlation between the feature maps of different channels as given by the Gram matrix (*Levy & Goldberg, 2014*) $G^{(l)}$. $G_{i,j}^{(l)}$ is the dot product of the features $i$, and $j$ in the layer $l$, it is which is given by:

$$G_{i,j}^{(l)} = vec\left(ch_i^{(l)}(I)\right).vec\left(ch_j^{(l)}(I)\right) \tag{23}$$

Where $vec()$ is the vectorization operation and $ch_i^{(l)}$ denoted the $i^{th}$ channel of feature maps in the layer $l$. Now the texture loss is given by (24)

$$\mathcal{L}_{TEX}\left(I,\hat{I};ch,l\right) = \frac{1}{c_l^2}\sqrt{\sum_{i,j}\left(G_{i,j}^{(l)}(I) - G_{i,j}^{(l)}(\hat{I})\right)^2} \tag{24}$$

Using the texture loss function in (24), EnhanceNet (*Sajjadi, Scholkopf & Hirsch, 2017*) reported more realistic results that look visually similar to the reference HR image. Although an optimized texture loss function-based SR generates more realistic-looking images, the selection of patch size is still an open field of research. The selection of small patch size leads to the generation of artifacts in the textured region, while selecting a big patch size generates artifacts across the whole image as the patches are averages over the whole image.

Total Variation Loss. Using the pixel values of the neighboring pixels, the total variation loss (*Rudin, Osher & Fatemi, 1992*) was defined as the sum of the absolute difference among the values of the neighboring pixels as:

$$\mathcal{L}_{TV}\left(\hat{I}\right) = \frac{1}{hwc}\sum_{i,j,k}\sqrt{\left(\hat{I}_{i+1,j,k} - \hat{I}_{i,j,k}\right)^2 - \left(\hat{I}_{i,j+1,k} - \hat{I}_{i,j,k}\right)^2} \tag{25}$$

Total variation loss was used in *Ledig et al. (2017)* and *Yuan et al. (2018)* to ensure smoothness across sharp edges/transitions within the generated image.

Cycle Consistency Loss. Using the CycleGAN (*Zhu et al., 2017a*) image SR method was presented in *Yuan et al. (2018)* using the cyclic consistency loss function. Using the generated HR image $\hat{I}$, the network generated another LR image $I'_{LR}$, which is further compared with the input LR image $I_{LR}$ for cyclic consistency.

In practice, various loss functions are used as a combination in SR to ensure various aspects of the generation process in the form of a weighted average as in *Kim, Lee & Lee (2016a)*, *Wang et al. (2018b)*, *Sajjadi, Scholkopf & Hirsch (2017)* and *Lai et al. (2017)*.

The selection of appropriate weights of the loss functions in itself is another learning problem as the results vary significantly by varying the weights of the loss function in image SR.

### Curriculum learning

In the Curriculum learning technique (*Bengio et al., 2009*), the method adapts itself to the variable difficulty of tasks, i.e., starting from simple images with minimum noise to complex images. Since SR always suffers from adverse conditions, the curriculum approach is mainly applied to its learning difficulty and network size. For reducing the training difficulty of the network in SR, small scaling factor, SR is performed in the beginning; in the curriculum learning-based SR, the training starts with 2× upsampling, and gradually the following scaling factors 4×, 8×, and so on are generated using the output of previously trained networks. ProSR (*Wang et al., 2018a*) uses the upsampled output of the previous level and linearly trains the next level using the previous one, while ADRSR (*Bei et al., 2018*) concatenates the HR output of the previous levels and further adds another convolution layer. In CARN (*Ahn, Kang & Sohn, 2018b*), the previously generated image is entirely replaced by the next level generated image, updating the HR image in sequential order.

Another alternative is to transform the image SR problem into N subsets and gradually solving these problems; as in *Park, Kim & Chun (2018)*, the 8× upsampling problem was divided into three problems (i.e., $1 \times$ to $2\times$; $2 \times$ to $4\times$ and $4 \times$ to $8\times$) and three separate networks were used to solve these problems. Using a combination of the previous reconstruction, the next level was finetuned in this method. The same concept was used in *Li et al. (2019b)* to train the network from low image degradations to high image degradations, thus gradually increasing the noise in the LR input image. Curriculum learning reduces the training difficulty; hence, the total computational time is also reduced.

### Batch normalization

Batch normalization (BN) was proposed by *Ioffe & Szegedy (2015)* to stabilize and accelerate the deep CNNs by reducing the internal covariate shift of the network. Every mini-batch was normalized, and two additional parameters were used per channel to preserve the representation ability. Batch normalization is responsible for working on the intermediate feature maps; thus, it resolves the vanishing gradient issue while allowing high learning rates. This technique is widely used in SR models such as *Ledig et al. (2017)*, *Zhang, Zuo & Zhang (2018)*, *Tai, Yang & Liu (2017)*, *Tai et al. (2017)*, *Ledig et al. (2017)*, *Sønderby et al. (2017)*, *Tai et al. (2017)*, *Tai, Yang & Liu (2017)*, *Liu et al. (2018b)* and *Zhang, Zuo & Zhang (2018)*. In contrast, (*Lim et al., 2017*) claimed that batch normalization-based networks lose the scale information of the generated images. Thus, there is a lack of flexibility in the network; hence, *Lim et al. (2017)* removed batch normalization and used the additional memory to design a large model with superior performance compared to the BN-based network. Other studies *Wang et al. (2019c)*, *Wang et al. (2018a)* and *Chen et al. (2018a)* also implemented this technique to achieve marginally better performance.

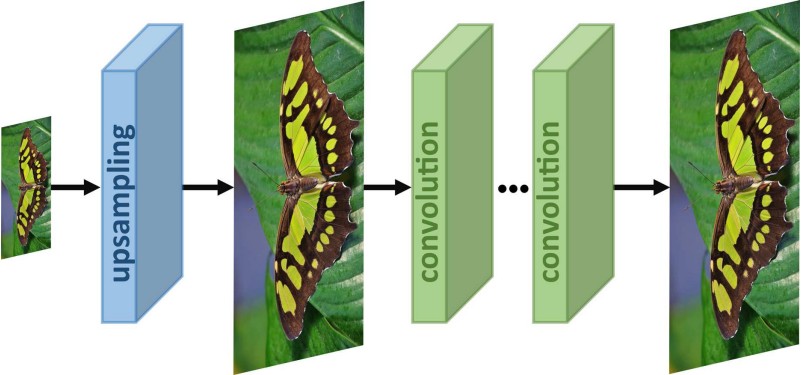

**Figure 7 Pre-upsampling-based super-resolution network pipeline.**

### Multi-supervision

Using numerous supervision signals within the same model for improving the gradient propagation and evading the exploding/vanishing gradient problem is called multi-supervision. In *Kim, Lee & Lee (2016b)*, multi-supervision is incorporated within the recursive units to address the gradient problems. In SR, the multi-supervision learning technique is implemented by catering to a few other factors in the loss function, which improves the back-propagation path and reduces the training difficulty of the model.

## SR frameworks

SR being an ill-posed problem; thus, upsampling is critical in defining the performance of the SR method. Based on learning strategies, upsampling methods, and network types, there are several frameworks for SR; here, four of them are discussed in detail, especially in light of the upsampling method used within the framework, as shown in Figs. 7–10.

### Pre-upsampling SR

Learning the mapping functions for upsampling from an LR image directly to an HR image is done using this framework, where the LR image is upsampled in the beginning, and various convolution layers are used to extract representations in an iterative way using deep neural networks. Using this concept *Dong et al. (2014, 2016)* introduced the pre-upsampling-based SR framework (SRCNN), as shown in Fig. 7. SRCNN was used to learn the end-to-end mapping of LR-HR image conversion using CNNs. Using the classical methods of upsampling as discussed in "Conventional Methods of Super-Resolution", the LR image is firstly converted to an HR image, and then deep CNNs were used to learn the representations for mapping the HR image.

Since the pre-upsampling layer already performs the actual pixel conversion task, the network needs to refine the results using CNNs; this results in reduced learning difficulty. Compared to single-scale SR (*Kim, Lee & Lee, 2016a*), which uses specific scales of input, these models can handle any random size image for refinement and have similar performance. In recent years, many application-oriented research studies have used this framework (*Kim, Lee & Lee, 2016b*; *Shocher, Cohen & Irani, 2018*; *Tai, Yang & Liu, 2017*;

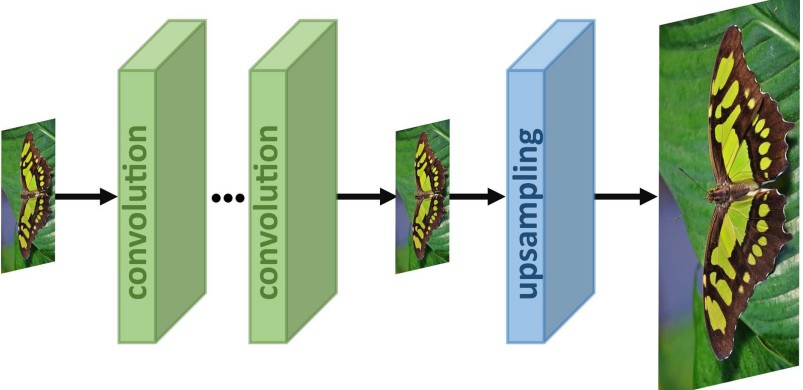

**Figure 8  Post-upsampling-based super-resolution network pipeline.**

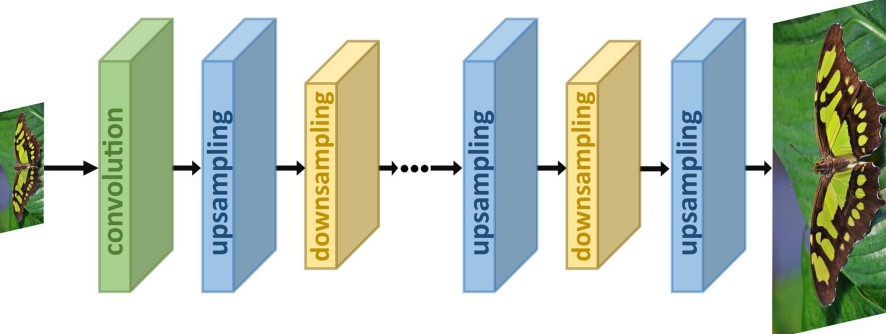

**Figure 9  Iterative up-and-down sampling-based super-resolution network pipeline.**

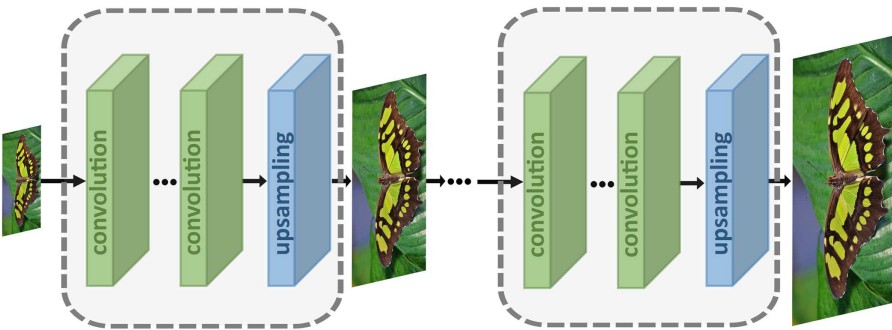

**Figure 10  Progressive sampling-based super-resolution network pipeline.**

_Tai et al., 2017_), the differences in these models are in the deep learning layers employed after the upsampling. The only drawback in this model is the use of a predefined classical method of pre-upsampling, which often results in the introduction of image blur, noise amplification in the upsampled image, which later affects the quality of the concluding HR image. Moreover, the dimensions of the image are increased at the start of

the method. Thus, the computational cost and memory requirements of this framework are higher (*Shi et al., 2016*).

### Post-upsampling SR

To minimize the memory requirements and increase computational efficiency, the post-upsampling method was used in SR to utilize deep learning to learn the mapping functions in low-dimensional space. This concept was first used in SR by *Shi et al. (2016)* and *Dong, Loy & Tang (2016)*, and the network diagram is shown in Fig. 8.

Due to low computational costs and the use of low-dimensional space for deep learning, this model has been widely used in SR because this reduces the complexity of the model (*Ledig et al., 2017*; *Lim et al., 2017*; *Tong et al., 2017*; *Han et al., 2018*).

### Iterative up-and-down sampling SR

Since the LR–HR mapping is an ill-posed problem, efficient learning using the LR-HR image pair using back-propagation (*Irani & Peleg, 1991*) was used in SR (*Timofte, Rothe & Van Gool, 2016*). The SR network is called the iterative up-down sampling SR, as shown in Fig. 9. This model refines the image using recursive back-propagation, i.e., continuously measuring the error and refining the model based on the reconstruction error. The DBPN method proposed in *Haris, Shakhnarovich & Ukita (2018)* used this concept to perform continuous upsampling and downsampling, and the final image was constructed using the intermediate generations of the HR image.

Similarly, SRFBN (*Li et al., 2019b*) used this technique with densely connected layers for image SR, while RBPN (*Haris, Shakhnarovich & Ukita, 2019*) used recurrent back-propagation with iterative up-down upsampling for video SR. This framework has shown significant improvement over the other frameworks; still, the back-propagation modules and their appropriate use require further exploration as this concept is recently introduced.

### Progressive-upsampling SR

Since the post-upsampling framework uses a single layer at the end of upsampling and the learning is fixed for scaling factors; thus, multi-scale SR will increase the computational cost of the post-upsampling framework. Thus, using progressive upsampling within the framework to gradually achieve the required scaling was proposed, as seen in Fig. 10. An example of this framework is the LapSRN (*Lai et al., 2017*), which uses cascaded CNN-based modules responsible for mapping a single scaling factor, and the output of one module acts as the input LR image to the other module. This framework was also used in ProSR (*Wang et al., 2018a*) and MS-LapSRN (*Lai et al., 2017*).

This model achieves higher learning rates as the learning difficulty is less since the SR operation is segregated into several small upscaling tasks, which is more straightforward for CNNs to learn. Furthermore, this model has built-in support for multi-scale SR as the images are scaled with various intermediate scaling factors. Training stability and convergence are the main issues with this framework, and this requires further research.

## Other improvements

Apart from the four primary considerations in image SR, other factors have a significant effect on the performance of a super-resolution method, and in this section, a few are discussed in light of recent research.

### Data augmentation

Data augmentation is a common technique in deep learning, and this concept is used to further enhance the performance of a deep learning model by generating more training data using the same dataset. In the case of image super-resolution, some of the augmentation techniques are flipping, cropping, angular rotation, skew, and color degradation (*Timofte, Rothe & Van Gool, 2016*; *Lai et al., 2017*; *Lim et al., 2017*; *Tai, Yang & Liu, 2017*; *Han et al., 2018*). Recoloring the image using channel shuffling in the LR-HR image pair is also used as data augmentation in image SR (*Bei et al., 2018*).

### Enhanced prediction

This data augmentation method affects the output HR image as multiple LR images are augmented using rotation and flipping functions (*Timofte, Rothe & Van Gool, 2016*). These augmented are fed to the model for reconstruction, the reconstructed outputs are inversely transformed, and the final HR image is based on the mean (*Timofte, Rothe & Van Gool, 2016*; *Wang et al., 2018a*) or median (*Shocher, Cohen & Irani, 2018*) pixel values of the corresponding augmented outputs.

### Network fusion and interpolation

This technique used multiple models to predict the HR image, and each prediction acts as the input to the following network, like in context-wise network fusion (CNF) (*Ren, El-Khamy & Lee, 2017*). The CNF was based on three individual SRCNNs, and this model achieved the performance, which was compared with the state-of-the-art SR models (*Ren, El-Khamy & Lee, 2017*).

In the SR network, network interpolation is a model that uses PSNR-based and GAN-based models for image SR to boost SR performance. Network interpolation strategy (*Wang et al., 2019c*, *2019b*) used a PSNR-based model for training. In contrast, a GAN-based model was used for fine-tuning while the parameters were interpolated to get the weights of interpolation, and their results had few artifacts and look realistic.

### Multi-task learning

Multi-task learning is used for learning various problems and getting a generalized model for representations found in learning, for example, image segmentation, object detection, and facial recognition (*Caruana, 1997*; *Collobert & Weston, 2008*). In the field of super-resolution, *Wang et al. (2018b)* used semantic maps as input to the model and predicted the parameters of the affine transformation on the transitional feature maps. The SFT-GAN in *Wang et al. (2018b)* generated more realistic and crisp-looking images with good visual details regarding the textured regions. While in DNSR (*Bei et al., 2018*), a denoising network was proposed to denoise the output generated by the SR network; thus, using this closed-loop system (*Bei et al., 2018*) was able to achieve good results. Like

DNSR, (*Yuan et al., 2018*) proposed an unsupervised SR using the cycle-in-cycle GAN (CinCGAN) for denoising during the SR task. Using a multi-tasking framework may increase the computational difficulty, but the system's performance is also enhanced in terms of PSNR and perceptual quality indexes.

## State-of-the-art SR methods

The recent year has excelled in developing SR models, especially using supervised deep learning; thus, the models have excelled in achieving state-of-the-art performance. Previously various aspects of the SR models and their underlying components were discussed in light of their strengths and weaknesses. In recent times the use of multiple learning strategies is common, and most of the state-of-the-art methods have used a combination of these strategies.

The first innovation was using the dual-branched network (DBCN) (*Gao, Zhang & Mou, 2019*) to increase the computational efficiency of the single-branched network by using a smaller number of convolutional layers for representation. Furthermore, in RCAN (*Zhang et al., 2018a*), attention-based learning was used combined with residual learning, L1 pixel loss function, and subpixel upsampling method to achieve the state-of-the-art results in image SR. Furthermore, various models and their reported results and some key factors are summarized in Table 5.

In previous sections, we discussed various strategies and compared and contrasted them; while these are important, the performance of any SR algorithm in comparison to the computational cost and parameters is also vital. In Fig. 11, we have graphically shown the performance of SR methods using PSNR metrics compared to their size (represented as several parameters) and computational cost (measured by the number of Multi-Adds). The datasets used in measurements are Set14, B100 and Urban 100; the overall PSNR is the average score over the three datasets, while the scaling factor for these models was fixed to $2\times$.

As evident in Fig. 11, the five best-performing methods on the selected datasets based on PSNR are WRAN (34.790 dB), RCAN (34.540 dB), SAN (34.480 dB), Meta-RDN (34.400 dB) and EDSR (34.330 dB). The variation in the average PSNR reported by these methods only varies in the range of 0.46 dB. There is a significant difference in the number of parameters reported by these five methods; WRAN reported only 2.710 million parameters while EDSR reported the highest parameters among the five methods, i.e., 40.74 million. WRAN and RCAN performed well in terms of PSNR, the number of parameters, and computational cost, thereby making them one of the best methods for image super-resolution.

## UNSUPERVISED SUPER-RESOLUTION

In this section, the methods of unsupervised SR are discussed, which does not require LR–HR pairs. The limitation of the supervised learning methods is that the LR images are usually generated using known degradations. In supervised learning, the model learns the reverse transformation function of the degradation function to convert the LR image into the HR image. Thus, using the unsupervised model to upsample the LR images is a

**Table 5 SR method details of various SR algorithms.**

| Year | Method name | US | Network | Framework | Loss function | Details |
|---|---|---|---|---|---|---|
| 2014, ECCV | SRCNN (*Dong et al., 2014*) | Bicubic | CNN | Pre | $\mathcal{L}_{L2}$ | First deep learning-based SR |
| 2016, CVPR | DRCN (*Kim, Lee & Lee, 2016b*) | Bicubic | Res., Rec. | Pre | $\mathcal{L}_{L2}$ | Recursive layers |
| 2016, ECCV | FSRCNN (*Dong et al., 2016*) | Deconv | | Post | $\mathcal{L}_{L2}$ | Lightweight |
| 2017, CVPR | ESPCN (*Caballero et al., 2017*) | Sub-pixel | | Pre | $\mathcal{L}_{L2}$ | Sub-pixel |
| 2017, CVPR | LapSRN (*Lai et al., 2017*) | Bicubic | Res. | Prog | $\mathcal{L}_{L1}$, $\mathcal{L}_{PIX\_CH}$ | Cascaded CNN |
| 2017, CVPR | DRRN (*Tai et al., 2017*a) | Bicubic | Res., Rec. | Pre | $\mathcal{L}_{L2}$ | Recursive layer blocks |
| 2017, CVPR | SRResNet (*Ledig et al., 2017*) | Sub-pixel | Res. | Post | $\mathcal{L}_{L2}$ | Content loss |
| 2017, CVPR | SRGAN (*Ledig et al., 2017*) | Sub-pixel | Res. | Post | $\mathcal{L}_{GAN}$ | GAN-based loss |
| 2017, CVPR | EDSR (*Lim et al., 2017*) | Sub-pixel | Res. | Post | $\mathcal{L}_{L1}$ | Compact design |
| 2017, ICCV | EnhanceNet (*Sajjadi, Scholkopf & Hirsch (2017)*) | Bicubic | Res. | Pre | $\mathcal{L}_{GAN}$ | GAN-based loss |
| 2017, ICCV | MemNet (*Tai et al., 2017*) | Bicubic | Res., Rec., Dense | Pre | $\mathcal{L}_{L2}$ | Memory layers blocks |
| 2017, ICCV | SRDenseNet (*Tong et al., 2017*) | Deconv | Res., Dense | Post | $\mathcal{L}_{L2}$ | Fully connected layers |
| 2018, CVPR | DBPN (*Haris, Shakhnarovich & Ukita, 2018*) | Deconv | Res., Dense | Iter | $\mathcal{L}_{L2}$ | Back-prop. Based |
| 2018, CVPR | DSRN (*Han et al., 2018*) | Deconv | Res., Rec. | Pre | $\mathcal{L}_{L2}$ | Dual-state network |
| 2018, CVPRW | ProSR, ProGanSR (Wang et al., 2018) | Progressive Upscale | Res., Dense | Prog | $\mathcal{L}_{LS}$ | Least square loss |
| 2018, ECCV | MSRN (*Li et al., 2018*) | Sub-pixel | Res. | Post | $\mathcal{L}_{L1}$ | Multi-path |
| 2018, ECCV | RCAN (*Zhang et al., 2018a*) | Sub-pixel | Res., Attent. | Post | $\mathcal{L}_{L1}$ | Attention-based loss |
| 2018, ECCV | ESRGAN (*Wang et al., 2019c*) | Sub-pixel | Res., Dense | Post | $\mathcal{L}_{L1}$ | GAN-based loss |
| 2019, CVPR | Meta-RDN (*Hu et al., 2019*) | Meta Upscale | Res., Dense | Post | $\mathcal{L}_{L1}$ | Multi-scale model |
| 2019, CVPR | Meta-SR (*Hu et al., 2019*) | Meta Upscale | Res., Dense | Post | $\mathcal{L}_{L1}$ | Arbitrary scale factor as input |
| 2019, CVPR | RBPN (*Haris, Shakhnarovich & Ukita, 2019*) | Sub-Pixel | Rec. | Post | $\mathcal{L}_{L1}$ | Used SISR and MISR together for VSR |
| 2019, CVPR | SAN (*Dai et al., 2019*) | Sub-Pixel | Res., Attent. | Post | $\mathcal{L}_{L1}$ | $2^{nd}$ order attention |
| 2019, CVPR | SRFBN (*Li et al., 2019b*) | Deconv | Res., Rec., Dense | Post | $\mathcal{L}_{L1}$ | Feedback path |
| 2020, Neuro-computing | WRAN (*Xue et al., 2020*) | Bicubic | Res., Attent. | Pre | $\mathcal{L}_{L1}$ | Wavelet-based |

**Note:**
"US," "Rec.," "Res.," "Attent.," "Dense," "Pre.," "Post.," "Iter.," and "Prog." represent upsampling methods, recursive learning, residual learning, attention-based learning, dense connections, pre-upsampling framework, post-upsampling framework, iterative up-down upsampling framework, and progressive upsampling framework respectively.

field of growing interest, where the model learns the real-world image degradation to achieve SR using the information of unpaired LR and HR images. A few of the unsupervised SR models are discussed in the sub-sections.

## Weakly-supervised super-resolution
The first method to address the use of known degradation in the model for the generation of LR images using weakly supervised deep learning, this method utilized the unpaired LR and HR images for training the model. Although this model still requires both LR and

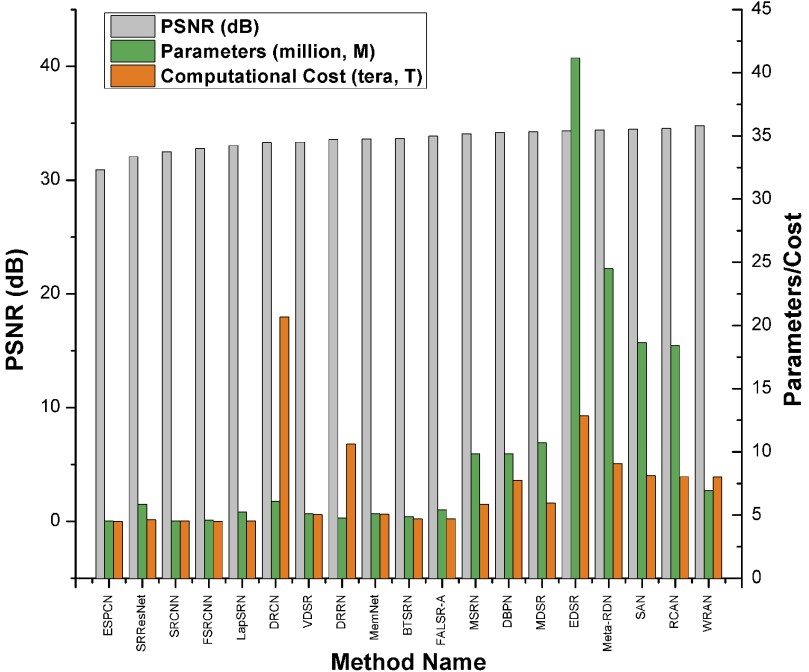

**Figure 11 Benchmarking of super-resolution models.** Image quality index is represented by PSNR (in *blue* color), which is a significant evaluation indicator of any super-resolution method; the total number of parameters learned by every method is shown in *green*. The computational efficiency is measured in tera multiply-adds, and it is shown in *orange* color.

HR images, the associations are not defined. Thus, there are two possible approaches; the first one is to learn the degradation function first, which can generate the degraded LR images and train the model to generate the HR images. The other method is to employ degradation function learning and LR-HR mapping cyclically, thus validating the results with each other (*Ignatov et al., 2018a*).

### Cyclic weakly-supervised SR

Using the unpaired LR and HR images and referring them to two separated uncorrelated datasets, this method uses a cycle-in-cycle approach to predict the mapping function of these two datasets, i.e., from LR to HR and HR to LR images. This is a recursive process where the mapping functions generate images with equal distribution, and these images are fed to the second prediction cyclically.

Using the deep learning-based CycleGAN (*Zhu et al., 2017a*), a cycle-in-cycle SR framework was proposed in *Yuan et al. (2018)* this framework used a total of four generators, while there were two discriminators; the two GANS learned the representation of degraded LR to LR and LR to HR mappings. In *Yuan et al. (2018)*, the first generator is a simple denoising element that generates similar scale denoised LR images; these denoised images act as input to the second generator to regenerate the HR image, which is further validated by the adversarial network, i.e., a discriminator. Thus, using different loss functions, the CycleGAN achieves image SR using weakly supervised learning.

Although this method has achieved comparable results, especially in very noisy images where classical degradation functions in supervised learning cannot be used, there is room for research to decrease the learning difficulty of the computational cost of this method.

### Learning the degradation function

A similar concept to the cyclic SR, but the two networks, i.e., a degradation learning network and LR-HR mapping network, are independently trained. In *Bulat, Yang & Tzimiropoulos (2018)*, a two-staged method of image SR was proposed, where a GAN learns the representations of the HR to LR transformation while the second GAN is trained using the paired output of the first GAN to learn the mapping representations of the LR to HR transformation. This two-stage model outperformed the state-of-the-art in Fréchet Inception Distance (FID) (*Heusel et al., 2017*) with 10% failure cases. This method reported superior reconstruction of HR human facial features.

## Zero-shot super-resolution

Using the training concept at the time of the test, the zero-shot SR (ZSSR) (*Shocher, Cohen & Irani, 2018*) uses a single image to train a deep learning network using image augmentation techniques to learn the degradation function. ZSSR was used (*Michaeli & Irani, 2013*) to predict the degradation kernel, which was further used to generate scaled and augmented images. The final step was to train an SRCNN network to learn the representations of this dataset, and in this way, the ZSSR uses augmentation and input image data to achieve SR. This model outdid the state-of-the-art for non-bicubic, noisy, and blurred LR images by 1dB in the case of estimated kernels and 2dB for known kernels.

Since this model requires training for every input image at the test time, the overall inference time is substantial.

## Image prior in SR

The low-level details in any learning problem can be mapped using CNNs, thus using a randomly initialized CNN as an image prior (*Ulyanov, Vedaldi & Lempitsky, 2020*) to perform SR. The network is not trained; instead, it uses a random vector $v$ as input to the model, and it generates the HR image $I_{yHR}$. This method aims to determine an image $\hat{I}_{yHR}$ that, when downsampled, returns an LR image that is similar to the input LR image $\hat{I}_{xHR}$. The model performed 2dB the state-of-the-art methods but reported superior results than the conventional bicubic upsampling method by 1dB.

## DOMAIN-SPECIFIC APPLICATIONS OF SUPER-RESOLUTION

In this section, various applications of SR grouped by the application domains are discussed.

## Face image super-resolution

Face hallucination (FH) is perhaps the ultimate target utility of the image SR for face-recognition-based tasks such as (*Gunturk et al., 2003*; *Taigman et al., 2014*; *Korshunova et al., 2017*; *Zhang et al., 2018c*; *Grm, Scheirer & Štruc, 2020*). The facial images contain

facial-structured information; thus, using image priors in FH has been a common approach to achieve FH.

Using techniques such as in CBN (*Zhu et al., 2016*), the generated HR images can be constrained to face-related features, forcing the model to output HR images containing facial features. In CBN, this was achieved by using a facial prior and a dense correspondence field estimation. While in FSRNet (*Chen et al., 2018b*) facial parsing maps and facial landmark heatmaps were used as priors to the learning network to achieve face image SR, SICNN (*Zhang et al., 2018c*) used a joint training approach to recover the real identity using a super-identity loss function. Super-FAN (*Bulat & Tzimiropoulos, 2018*) approached FH using end-to-end learning with FAN to ensure the generated images are consistent with human facial features.

Using implicit methods for solving the face misalignment problem is another way to approach FH; for instance, in *Yu & Porikli (2017)*, the spatial transformation is achieved using transformation networks (*Jaderberg et al., 2015*). Another method based on (*Jaderberg et al., 2015*) is TDAE (*Wang et al., 2019b*), which uses a three-module approach for FH, using a D-E-D (decoder-encoder-decoder) model to achieve FH; the first decoder performs denoising and upsampling while encoder downsamples the denoised image which is fed to the final decoder for FH. Another approach is to use HR exemplars from datasets to decompose the facial features of an LR image and project the HR features into the exemplar dataset to achieve FH (*Yang, Liu & Yang, 2018*). In *Song et al. (2018)*, an adversarial discriminative network was proposed for feature learning on both feature space and raw pixel space; this method performed well for heterogenous face recognition (HFR).

In other research studies, human perception of attention shifting (*Najemnik & Geisler, 2005*) was used in Attention-FH (*Cao et al., 2017a*) to learn face patches for local enhancement of FH. In *Xu et al. (2017)*, a multi-class GAN network was proposed for FH, which composed of multiple generators and discriminators, while in *Yu & Porikli (2016)*, the authors adopted a network model analogous to SRGAN (*Ledig et al., 2017*). Using conditional GAN (*Gauthier, 2014*), the studies (*Lee et al., 2018b*; *Yu et al., 2018*) used additional facial features to achieve FH with predefined attributes. Gao et al. proposed an efficient multilayer locality-constrained matrix regression (MLCMR) framework for face super-resolution of highly degraded LR images (*Gao et al., 2021*).

## Real-world image super-resolution

In real-world images, the sensors used to capture them already introduce degradations as the final RGB (8-bit) image is converted from the raw image (usually more than 14-bit or higher). Thus, using these images as a reference for SR is not optimal as the images have already been degraded (*Wang, Chen & Hoi, 2020*). To approach this problem, research studies such as *Zhang et al. (2019a)* and *Chen et al. (2019)* have proposed methods for developing real-world image datasets. In *Zhang et al. (2019a)*, the SR-RAW dataset was developed by the authors, which contained raw-HR-LR(RGB) pairs generated using the optical zoom in cameras, while in *Chen et al. (2019)*, image resolution and its relationship with the field of view (FoV) were explored by the authors to generate a real-world dataset called City100.

## Depth map super-resolution

In the field of computer vision, problems like image segmentation (*Zaitoun & Aqel, 2015*; *Yu & Koltun, 2016*; *Kirillov et al., 2019*) and pose estimation (*Wei et al., 2016*; *Cao et al., 2017b*; *Chen & Ramanan, 2017*) have been approached by using depth maps. Depth maps retain the distance information of the scene and the observer, although these depth maps are of low-resolution because of the hardware constraints of the modern camera systems. Thus, image SR is used in this regard to increase the resolution of the depth maps.

Using multiple cameras to record the same scene and generate multiple HR images is the most suitable way of doing depth map SR. In *Hui, Loy & Tang (2016)*, the authors used two separate CNNs to downsample HR image concurrently and upsample the LR depth map; after the generation of RGB features from the downsampling CNN, these features were used to fine-tune the upsampling process of depth maps, while *Riegler, Rüther & Bischof (2016)* used the energy minimization model (such as *Bashir & Ghouri (2014)*) to guide the model for generating HR depth maps without the need for reference images.

## Remote sensing and satellite imaging

The use of SR in improving the resolution of remote sensing and satellite imaging has increased in the past years (*Shermeyer & Van Etten, 2019*). In *Li et al. (2017b)*, the authors used the concept of multi-line cameras to utilize multiple LR images to generate a high-quality HR image from the ZY-3 (TLC) satellite image dataset. In *Zhu et al. (2017b)* and *Benecki et al. (2018)*, the authors argued that the conventional methods of evaluation of the SR techniques are not valid for satellite imaging as the degradation functions and operation conditions of the satellite hardware are entirely in a different environment and thus (*Benecki et al., 2018*) proposed a new way for validation of SR methods for satellite image SR methods. An adaptive multi-scale detail enhancement (AMDE-SR) was proposed in (*Zhu et al., 2018*) to use the multi-scale SR method to generate high-detailed HR images with accurate textual and high-frequency information. GAN-based methods provide superior performance for remote sensing image SR; Liu et al. developed a novel cascaded conditional Wasserstein generative adversarial network (CCWGAN) to generate HR images for remote sensing (*Liu et al., 2020a*). Bashir et al. proposed a YOLOv3-based small-object detection framework SRCGAN-RFA-YOLO (*Bashir & Wang, 2021b*), where the authors used residual feature aggregation and cyclic GAN to improve the resolution of remote sensing images before performing object detection.

## Video super-resolution

In video SR, multiple frames represent the same scene; thus, there is inter and intra-frame spatial dependency in the video, which includes the information of brightness, colors, and relative motion of objects. Using the optical flow-based method (*Sun, Roth & Black, 2010*; *Liao et al., 2015*), Sun et al. and Liao et al. proposed a method to generate probable HR candidate images and ensemble these images using CNNs. Using the Druleas (*Drulea & Nedevschi, 2011*) algorithm, CVSRnet (*Kappeler et al., 2016*) addressed the effect of motion by using CNNs for the images in successive frames to generate HR images.

Apart from direct learning motion compensation, a trainable spatial transformer (*Jaderberg et al., 2015*) was used in VESPCN (*Caballero et al., 2017*) to motion compensation mapping using data from successive frames for end-to-end mapping. Using a sub-pixel layer-based module, (*Tao et al., 2017*) achieved super-resolution and motion compensation simultaneously.

Another approach is to use recurrent networks to indirectly grasp the spatial and temporal interdependency to address the motion compensation. In STCN (*Guo & Chao, 2017*), the authors used a bidirectional LSTM (*Graves, Fernández & Schmidhuber, 2005*) and deep CNNs to extract the temporal and spatial information from the video frames, while BRCN (*Huang, Wang & Wang, 2015*) utilized RNNs, CNNs, and conditional CNNs respectively for temporal, spatial and temporal-spatial interdependency mapping. Using 3D convolution filters of small size to replace the large-sized filter, FSTRN (*Li et al., 2019a*) achieves state-of-the-art performance using deep CNNs, sustaining a low computational cost. A novel spatio-temporal matching network (STMN) for video SR was proposed, which worked on the wavelet transform to minimize the dependence on motion estimations (*Zhu et al., 2021*).

## SR for medical imaging

Other fields also used the concept of image super-resolution to achieve high-resolution images, such as in *Mahapatra, Bozorgtabar & Garnavi (2019)*; the authors proposed the use of progressive GANs to enhance the image quality of magnetic resonance (MR) images. DeepResolve (*Chaudhari et al., 2018*) used image SR methods to generate thin-sliced knee MR images from the thick-sliced input images. Since the diffusion MRI has high image acquisition time and low resolution, Super-resolution Reconstruction Diffusion Tensor Imaging (SRR-DTI) reconstructed HR diffusion parameters from LR diffusion-weighted (DW) images (*Van Steenkiste et al., 2016*). *Hamaide et al. (2017)* also used SRR-DTI to find the structural sex variances in the adult zebra finch brain.

Assisted diagnosis using super-resolution has been a recent trend; for instance, researchers used deep learning-based SR methods to assist the diagnosis of movement disorders like isolated dystonia (*Bashir & Wang, 2021a*).

## Other applications

Other applications of SR include object detection (*Li et al., 2017a*; *Tan, Yan & Bare, 2018*), stereo image SR (*Duan & Xiao, 2019*; *Guo, Chen & Huang, 2019*; *Wang et al., 2019a*), and super-resolution in optical microscopy (*Qiao et al., 2021*). Overall, SR plays a vital role in multi-disciplines, from medical science, computer vision to satellite imaging and remote sensing.

## DISCUSSION AND FUTURE DIRECTIONS

This paper gives an overall review of literature for image super-resolution, and the contribution of this paper is discussed in this section.

## Learning strategies

Learning strategies in image SR are introduced in "Learning Strategies"; while the learning strategies are well matured in image SR, there are research directions in the development of alternate loss functions and alternative of batch normalization

There are various loss functions in SR, and the choice of SR depends upon the task, while it is still an open research area to find an optimal loss function that fits all SR frameworks. A combination of loss functions is currently used to optimize the learning process, and there are no standard criteria for the selection of loss function; thus, exploring various probable loss functions for super-resolution is a promising future direction.

Batch normalization is a technique that performs well in computer vision tasks and reduces the overall runtime of the training, and enhances the performance; however, in SR batch normalization proved to be sub-optimal (*Lim et al., 2017*; *Wang et al., 2018a*; *Chen et al., 2018a*). In this regard, normalization techniques for super-resolution should be explored further.

## Network design

Network design strategies require further exploration in SR as the network design inherently dictates the overall performance of any SR method. Some of the key research areas are highlighted in this section

As discussed in "Upsampling Methods", current upsampling methods have significant drawbacks for the deconvolution layer and may produce checkerboard artifacts. In contrast, the sub-pixel layer is susceptible to the non-uniform distribution of receptive fields; the meta-scale method has stability issues, while the interpolation-based methods lack end-to-end learning. Thus, further research is required to explore upsampling methods that can be generic to SR models and can be applied to LR images with any scaling factors.

For human perception in SR, further research is required in attention-based SR, where the models may be trained to give more attention to some image features than others like the human visual system does.

Using a combination of low and high-level representations simultaneously to accelerate the SR process is another field in network design for fast and accurate reconstruction of the HR image.

Exploring network architectures that can be implemented in practical applications since current methods use deep neural networks, which increases the performance of the SR at the expense of higher computational cost; thus, research in the development of network architecture that is minimal and provides optimal performance is another promising research direction.

## Evaluation metrics

The image quality metrics used in SR act as the benchmark score, while the two most commonly used metrics, PSNR and SSIM, help gauge the performance of SR, but these metrics introduce inherent issues in the generated image. Using PSNR as an evaluation metric usually introduces non-realistic smooth surfaces, while SSIM works with textures,

structures, brightness, and contrast to imitate human perception. These metrics cannot completely grasp the perceptual quality of images (*Ledig et al., 2017*; *Sajjadi, Scholkopf & Hirsch, 2017*). Opinion scoring is a metric that ensures perceptual quality, but this metric is impractical for implementing SR methods for large datasets; thus, a probable research direction is developing a universal quality metric for SR.

### Unsupervised super-resolution

In the past 2 years, unsupervised SR methods have gained popularity, but still, the task of collecting various resolution scenes for a similar pose is difficult; thus, bicubic interpolation is used instead to generate an unpaired SR dataset. In actuality, the unsupervised SR methods learn the inverse mapping of this interpolation for the reconstruction of HR images, and the actual learning of SR is still an open research field using unsupervised learning methods.

## CONCLUSION

A detailed survey of classical SR and recent advances in SR with deep learning are explored in this survey paper. The central theme of this survey was to discuss deep learning-based SR techniques and the application of SR in various fields. Although image SR has achieved a lot in the last decade, some open problems are highlighted in "Discussion and Future Directions". This survey is intended for the researchers in the field of SR and researchers from other fields to use image SR in their respective fields of interest.

## ACKNOWLEDGEMENTS

We show our gratitude to the authors of all referred research studies for sharing results, especially to the authors of *Kim, Lee & Lee (2016b)*, *Ledig et al. (2017)*, *Dong, Loy & Tang (2016)*, *Lai et al. (2017)*, *Haris, Shakhnarovich & Ukita (2018)*, *Hu et al. (2019)*, *Tai, Yang & Liu (2017)*, *Tai et al. (2017)*, *Li et al. (2018)*, *Lim et al. (2017)*, *Zhang et al. (2018a)*, *Dai et al. (2019)*, *Xue et al. (2020)* and *Caballero et al. (2017)*.

### Funding

This work was supported by the National Natural Science Foundation of China (No. 62071384) and the Natural Science Basic Research Plan in Shaanxi Province of China (No. 2019JM-311). The funders had no role in study design, data collection and analysis, decision to publish, or preparation of the manuscript.

### Grant Disclosures

The following grant information was disclosed by the authors:
National Natural Science Foundation of China: 62071384.
Natural Science Basic Research Plan in Shaanxi Province of China: 2019JM-311.

### Competing Interests

The authors declare that they have no competing interests.

## Author Contributions

- Syed Muhammad Arsalan Bashir conceived and designed the experiments, performed the experiments, analyzed the data, performed the computation work, prepared figures and/or tables, authored or reviewed drafts of the paper, and approved the final draft.
- Yi Wang analyzed the data, performed the computation work, authored or reviewed drafts of the paper, and approved the final draft.
- Mahrukh Khan performed the experiments, analyzed the data, performed the computation work, prepared figures and/or tables, authored or reviewed drafts of the paper, and approved the final draft.
- Yilong Niu analyzed the data, authored or reviewed drafts of the paper, and approved the final draft.

## Data Availability

The raw data for Figure 11 are available in the Supplemental File.

## Supplemental Information

Supplemental information for this article can be found online at http://dx.doi.org/10.7717/peerj-cs.621#supplemental-information.

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
