# Peer review of "Retraction Notice"

_PeerJ Computer Science, doi:10.7717/peerj-cs.621_

## Round 0.1 · original submission · Major Revisions

The manuscript received mixed review comments from 4 reviewers. I still have interest in this manuscript and would like to invite the authors for revising their manuscript. Authors are highly recommended to consider the following comments as well as the comments from the anonymous reviewers.

1. Some keywords are not emphasized in the Abstract.

2. In almost all equations, additional parentheses appear in the PDF in PDF version, please carefully check them.

3. Also, many mathematical symbols of the Equations shall be italicized in the text. Similarly, so many superscripts and subscripts are not properly written in the context.

4. I strongly recommend the authors consider adding recent studies on SR in the medical imaging field (in Section 7) since it is quite important and also this survey is not focused on specific types of images.

5. I do not think it is advisable to add Subsections 8.1-8.5 in the conclusions. Instead, Discussion sections will be much better to discuss these points with further elaboration.

6. There are a lot of studies that were recently published. Please consider the recent papers rather than old ones in literature to enrich this survey. Some examples are given below.

- https://dx.doi.org/10.1109/ACCESS.2019.2959940
- https://doi.org/10.1016/j.patcog.2020.107619
- https://doi.org/10.1038/s41592-020-01048-5
- https://doi.org/10.1007/s10489-020-02116-1
- https://doi.org/10.1016/j.patcog.2020.107798
- https://arxiv.org/abs/2011.04566
- https://arxiv.org/abs/2008.02382
- https://dx.doi.org/10.1109/ICICSP50920.2020.9232066
- https://doi.org/10.1038/s41592-021-01080-z
- https://doi.org/10.1016/j.patcog.2020.107539
- https://dx.doi.org/10.1109/ACCESS.2020.3024164

7. Finally, writing needs massive improvements by avoiding very long sentences, proper use of punctuation, and removing the abbreviations that are not frequently used in the context.

·

Basic reporting

STRONG ASPECTS
-Concise and readable abstract.
-Adequately mentions the contribution.

WEAK ASPECTS
-Ambiguity in readability and connectivity because of lots of sub-sections
-Figure 1 should mention the color-coding.

Experimental design

STRONG ASPECTS
-Categorization of SR methods seems to add value in the research domain
-Detailed tables and figures for defining the inclusion and exclusion criteria and datasets

WEAK ASPECTS
-Research could have considered more paired LR-HR image datasets described in the table
3
-Survey Methodology section 3, mentions the study covers papers from 2008-2021
whereas the next section named Conventional Methods for Super-Resolution i.e.
section 4 refers to studies and methods from 1971 and onwards.

Validity of the findings

STRONG ASPECTS
- The research states all the learning processes used in ML paradigm which makes it
highly recommended for the different novel ML scenarios
- Up-to-date literature and methodologies presented which would be relevant over years

WEAK ASPECTS
- Figure 11 should be discussed in detail.

Additional comments

A thorough review of all the existing methodologies representing their mathematical basis. Goodluck!

Reviewer 2 ·

Basic reporting

Although the writing in this paper is good, where it is difficult to find grammatical errors or typos, the mathematical symbols used need to be improved. Three main comments on mathematical symbols in this paper:
1) Some of the symbols are strange, mostly because the original font style is not embedded into the pdf file.
2) The symbols used in the equation, should be exactly the same with the one use in the text. For example, in equation (5), symbol L is used, but in text (i.e., on line 137), different symbol is been used.
3) Mathematical symbols in the text should be written in Italic.

The title of the paper limits the scope of the review to only for single image super-resolution methods. However, this paper failed to introduce the background of single image super-resolution, such as what is its definition, and how it differs from super-resolution from multiple images or video input. Thus, the author should give some background in this aspect in Section 1.

The paper is well organized, with related and nice figures provided. Table 5 has been provided to compare the structure of the methods. Yet, in my opinion, it would be much better if the authors could also provide the tables to summarize each section of this review. For example, at the end of Section 2.2., a summary section is provided to compare the quality measures for super-resolution. Then, in this section, better to provide a table to compare those quality measures, such as what are the advantages and disadvantages of PSNR, MSE, SSIM, etc.

The review not only covers a broad approaches for deep-learning based super-resolution methods, but it also reviews a broad possible applications for super-resolution. In my opinion, this will attract the attention and interest of the readers.

In line 66 to 75, the authors mention that there is no literature review on super-resolution using deep-learning approaches. However, this is not true. There are already many available literature review in this field, for examples:
1) W. Yang, X. Zhang, Y. Tian, W. Wang, J. Xue and Q. Liao, "Deep Learning for Single Image Super-Resolution: A Brief Review," in IEEE Transactions on Multimedia, vol. 21, no. 12, pp. 3106-3121, Dec. 2019, doi: 10.1109/TMM.2019.2919431.
2) L. Zhou and S. Feng, "A Review of Deep Learning for Single Image Super-Resolution," 2019 International Conference on Intelligent Informatics and Biomedical Sciences (ICIIBMS), 2019, pp. 139-142, doi: 10.1109/ICIIBMS46890.2019.8991477.
3) Ooi, Y.K.; Ibrahim, H. Deep Learning Algorithms for Single Image Super-Resolution: A Systematic Review. Electronics 2021, 10, 867. https://doi.org/10.3390/electronics10070867
4) Viet Khanh Ha, Jin-Chang Ren, Xin-Ying Xu, Sophia Zhao, Gang Xie, Valentin Masero and Amir Hussain. Deep Learning Based Single Image Super-resolution: A Survey. International Journal of Automation and Computing, vol. 16, no. 4, pp. 413-426, 2019 doi: 10.1007/s11633-019-1183-x
5) Zhang, H.; Wang, P.; Zhang, C.; Jiang, Z. A Comparable Study of CNN-Based Single Image Super-Resolution for Space-Based Imaging Sensors. Sensors 2019, 19, 3234. https://doi.org/10.3390/s19143234
Therefore, it is better if the authors could highlight how this review paper differs from the other available reviews.

Experimental design

This review paper is within the scope of PeerJ Computer Science. The review is organized.

Section 3 presents the survey methodology. The terms used for the search are listed in line 343 to 344, and Table 4 presents the inclusion and exclusion criteria. However, based on these descriptions, there is no criteria set for "single-image" for super-resolution. Thus, the results from the search may not limited for single-image approaches.

Validity of the findings

As stated in Part 1, it would be better if the authors could provide a summary at the end of each section, and provide a table to compare the reviewed approaches in terms of their advantages and disadvantages.

Additional comments

In addition to the comments in Parts 1-3, the following are the minor concerns regarding to the paper:

1) In line 36, the author mention that "The image-based computer graphics models lack of resolution independence". However, this statement is not true. Computer graphic is not necessarily in raster format (i.e., bitmap format), but may also in generated as a vector graphic. The vector graphic is resolution independent.

2) In line 48, the author mention "shows the importance of this topic in recent years". However, for the surveillance (in line 47), the references are from year 2003 and 2010, which are not recent. Better to provide newer references.

3) In line 91, the sentence is referring to Figure 1. However, no elaboration on Figure 1 is provided. Please give some elaboration on Figure 1 in this paragraph.

4) In line 399, "explored in Section 3". It is a wrong section. Please provide the correct section.

5) Why the structures for "Attention-based learning" (section 5.2.6) and "Wavelet transform based learning" (section 5.2.7) are not included in Figure 6?

6) In line 808, "used after the removal of batch normalization used the additional memory to design a large model" is confusing. Better to rephrase.

Reviewer 3 ·

Basic reporting

No comment.

Experimental design

No comment.

Validity of the findings

No comment.

Additional comments

In this study, Bashir et al. conducted a literature review on deep learning-based single image super-resolution. In my opinion, the review is comprehensive and fits the scope of the journal. However, it is a well-known study and some previous studies have reviewed it before i.e., https://doi.org/10.1007/s11633-019-1183-x, or https://doi.org/10.1007/s11554-019-00925-3, ... Therefore, the idea to have one more survey is not necessary maybe, and then the contribution of the study is limited.

The authors should explain more on the keyword query and collected data.

Why did the authors need to query ScienceDirect and IEEE Xplore since Google Scholar already covers these two?

There still have grammatical errors and typos. For example, even in Fig. 3: "Scient Direct".

Deep learning is well-known and thus, in its description, the authors should mention more deep learning-based studies in different fields such as PMID: 33816830 and PMID: 31750297.

What is "initial screening"? The authors should describe this part clearly.

·

Basic reporting

The review paper was comprehensive and specific to the chosen area of review

Experimental design

Not applicable. This is a review paper and its study design is limited in scope to reviewing articles in the given field. The given review article meets the aims and scope of the journal. This review includes sufficient details and information needed for a

Validity of the findings

The review paper summarizes the recent researches and their significance in the Image processing field. Its a needed review compilation and the first of its kind in the niche area

---

## Round 0.2 · Minor Revisions

The reviewers and I are almost satisfied with the revision except few minor comments before finally recommending the Accept decision.

Please, consider the comments of the 1st and the 3rd reviewer as well as the following.

- Subsection 7.6 needs to be improved. "SRR-DTI" and "DW" are not defined.
- Subsection 8.5 has no additional value, either improve or delete it.
- In Conclusion, the following sentence is out of context, please check.
"In this section, we summarize the future trends in SR."

·

Basic reporting

STRONG ASPECTS
- OK

WEAK ASPECTS
-Authors have not highlighted the significance of their study as compared to previous similar studies in the literature
- References need to be updated.

Experimental design

STRONG ASPECTS
-Categorization of SR methods seems to add value in the research domain
-Detailed tables and figures for defining the inclusion and exclusion criteria and datasets

Validity of the findings

STRONG ASPECTS
- The research states all the learning processes used in the ML paradigm
- Up-to-date literature and methodologies presented which would be relevant over years

WEAK ASPECTS
- Similar studies are available in the literature. The reader can not determine what additional value to the body of existing knowledge is increased by this research.

Reviewer 2 ·

Basic reporting

No comment

Experimental design

No comment

Validity of the findings

No comment

Additional comments

I am satisfied with the improvements done by the authors.

Reviewer 3 ·

Basic reporting

No comment.

Experimental design

No comment.

Validity of the findings

No comment.

Additional comments

Thanks for addressing my previous comments. There are some comments that still need room for improvement:
1. How did the authors remove the duplicate samples among ScienceDirect, IEEE Xplore, and Google Scholar since they must have overlap?
2. I suggested the authors cite more works related to deep learning-based medical works i.e., PMID: 33816830 and PMID: 31750297, but they skipped it.
3. Quality of figures should be improved.

·

Basic reporting

The authors have diligently worked on the comments the reviewers had on the reporting structure in their new revision

Experimental design

This review work is in compliance with the aims and scope of our esteemed journal

Validity of the findings

This review paper encompasses the gaps which were discussed with the authors and they have come back with the mentioned corrections

Additional comments

Appreciate your nimbleness in doing the needed changes and getting back this fast

---

## Round 0.3 · accepted · Accept

The manuscript has been improved and reviewers' comments have been addressed through the 2 rounds of revision. I believe it is acceptable for publication.